# PRUNE TO FIT: ENABLING FEDERATED FINE-TUNING WITHIN EDGE MEMORY BUDGETS

## ABSTRACT

Federated fine-tuning enables privacy-preserving Large Language Model (LLM) adaptation, but its high memory cost limits participation from resource-constrained devices. We propose FEDPRUNER, an innovative federated fine-tuning paradigm that tackles this via intelligent layer pruning. FEDPRUNER flexibly prunes the global model, creating personalized submodels based on device memory constraints. It employs a macro-micro synergistic pruning framework: a macro-level functionality-driven layer orchestration mechanism groups layers, while a micro-level importance-aware layer selection strategy prunes within groups to build device-specific submodels. We further introduce a fine-grained variant that independently prunes Multi-Head Attention and Feed-Forward Network components to precisely preserve critical architectural elements. Extensive experiments demonstrate that FEDPRUNER significantly outperforms state-of-the-art methods with average accuracy gains of up to 11.11%. Moreover, it maintains strong robustness under varying memory constraints, yielding a 1.98% average performance improvement while reducing peak memory usage by 75%. [1]

## 1 INTRODUCTION

Large Language Models (LLMs) (Zhao et al., 2023; Minaee et al., 2024) have demonstrated remarkable performance across a wide range of tasks. However, fine-tuning pre-trained LLMs for downstream tasks necessitates a significant amount of task-specific data, which is hard to obtain due to privacy concerns (Wang et al., 2024). Federated fine-tuning (Zhang et al., 2024a) has emerged as a promising solution, enabling the adaptation of LLMs while preserving data privacy. In response to the prohibitive nature of full parameter fine-tuning on resource-constrained devices, researchers have proposed various parameter-efficient federated fine-tuning approaches, with Low-Rank Adaptation (LoRA) (Hu et al., 2021) distinguishing itself through its exceptional efficiency and flexibility (Guo et al., 2024).

Despite these advantages, federated fine-tuning with LoRA remains challenged by the high memory requirements of LLMs (Wu et al., 2025b), which dramatically outstrip the memory resources available in edge devices. Following Fed-pilot (Zhang et al., 2024c), the total memory consumption can be formulated as Equation 1, where $M_{\text{params}}$, $M_{\text{optimizer}}$, $M_{\text{activations}}$, and $M_{\text{system}}$ denote the memory occupied by model parameters, optimizer states, intermediate activations, and system/CUDA context, respectively. Given that system overhead is highly device-dependent and model-intrinsic memory dominates the total footprint, this work primarily targets the optimization of the latter. In the context of LLMs, this results in a prohibitive resource gap: for instance, fine-tuning LLaMA2-7B (Touvron et al., 2023) demands up to 26.9 GB of memory, whereas off-the-shelf edge devices typically possess only 4-12 GB (Xu et al., 2023; Tam et al., 2024).

$$M_{\text{total}} = \underbrace{M_{\text{params}} + M_{\text{optimizer}} + M_{\text{activations}}}_{\text{Model-Intrinsic Memory}} + \underbrace{M_{\text{system}}}_{\text{System Memory}} \tag{1}$$

To address the memory constraints, various approaches have been proposed, falling into two categories: 1) *Rank Heterogeneity*, where devices are assigned different ranks based on their available

---

[1]Code of FEDPRUNER is submitted for reproducibility and will be publicly available.

resources (Cho et al., 2024; Wang et al., 2024), thus reducing the trainable parameters in LoRA modules, and 2) *Zeroth-Order Optimization*, which only performs forward propagation to update model parameters, removing the need to store intermediate activations (Xu et al., 2023; Qin et al., 2023). However, these methods yield limited memory savings, as both LoRA modules and activations constitute only a small fraction of the total memory footprint, with model parameters consuming the majority of the memory. For example, when fine-tuning LLaMA2-7B, model parameters occupy 92.8% of memory, while LoRA modules and activations account for only 0.018% and 7.2%, respectively. Therefore, compared to optimizing LoRA modules and activations, *reducing model parameters is more promising to lower training memory usage*.

Building on this insight that model parameters dominate the memory footprint, we propose FED-PRUNER, an innovative federated fine-tuning paradigm that addresses memory constraints via intelligent layer pruning. Specifically, we strategically prune the layers of the global model, reducing model parameters to accommodate device memory constraints. However, a critical challenge lies in:

> *How to coordinate the pruning process across devices to optimize model performance and training efficiency under resource constraints?*

To address this challenge, we develop a macro-micro synergistic pruning framework that jointly considers layer functional characteristics and importance to coordinate the pruning process. Specifically, at the macro level, we propose a functionality-driven layer orchestration mechanism that adaptively organizes layers into functional groups based on their roles in data processing. For each group, we select one layer for submodel construction while pruning the remaining layers, thereby guaranteeing that the resulting submodel maintains functional completeness and hierarchical information processing capabilities. Moreover, to select the most representative layer from each group, we propose a micro-level importance-aware layer selection strategy where the preservation probability of each layer correlates with its contribution to model performance. This strategy effectively maintains critical computational pathways and accelerates model convergence. Furthermore, to enable more flexible submodel construction, we propose FEDPRUNER$^+$, a fine-grained pruning framework that performs pruning at the component level. Overall, our key contributions are summarized as follows:

- We introduce FEDPRUNER that comprehensively explores how to intelligently perform layer pruning to address the memory constraints in federated fine-tuning.
- We develop a macro-micro synergistic pruning framework to coordinate the pruning process across devices. Moreover, we propose a fine-grained variant that enables more precise pruning control at the component level.
- We conduct extensive experiments to demonstrate the effectiveness of FEDPRUNER. The results show that it achieves up to **11.11%** higher average accuracy than existing methods, while maintaining robustness under diverse memory constraints.

## 2 PRELIMINARY AND MOTIVATION

### 2.1 BASICS OF LoRA

The core idea of LoRA (Hu et al., 2021) involves keeping the pre-trained weight matrix $\mathbf{W}_0 \in \mathbb{R}^{U_1 \times U_2}$ frozen while parameterizing its update $\mathbf{\Delta W}$ via low-rank factorization. Specifically, $\mathbf{\Delta W}$ is decomposed into the product of two trainable matrices $\mathbf{B} \in \mathbb{R}^{U_1 \times r}$ and $\mathbf{A} \in \mathbb{R}^{r \times U_2}$, with $r \ll \min(U_1, U_2)$. The input is processed in parallel by $\mathbf{W}_0$ and $\mathbf{\Delta W}$, and their outputs are merged through element-wise addition. For a linear layer $h = \mathbf{W}_0 x$, the modified forward propagation is formulated as: $h = \mathbf{W}_0 x + \mathbf{BA} x$.

### 2.2 MEMORY WALL IN FEDERATED FINE-TUNING

We investigate the feasibility of deploying federated fine-tuning on edge devices, focusing on memory requirements. Profiling reveals a significant "memory wall." As shown in Figure 1(a), fine-tuning even smaller models like TinyLLaMA with LoRA requires 5.4 GB of memory. This requirement escalates sharply for larger models such as LLaMA3-3B (14 GB) and LLaMA2-7B (26.9 GB), far exceeding the maximum available memory of 12 GB on edge devices (Tam et al., 2024). Consequently, devices lack sufficient memory to perform local fine-tuning, hindering federated learning.

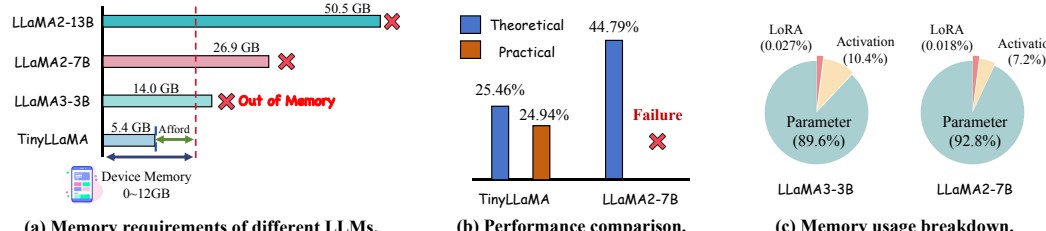

(a) Memory requirements of different LLMs.  (b) Performance comparison.  (c) Memory usage breakdown.

Figure 1: The memory wall in federated fine-tuning. Note that system-related memory overhead is excluded from this analysis. (a) Memory requirements for fine-tuning various LLMs using LoRA. Measurements are based on unquantized model parameters with a batch size of 16 and a maximum sequence length of 512. (b) Performance comparison between practical deployment and the theoretical baseline on MMLU. (c) Detailed breakdown of memory usage.

Moreover, this memory constraint severely impacts model performance in practical deployments. Figure 1(b) demonstrates a significant performance drop compared to the theoretical scenario without memory limits. For TinyLLaMA, MMLU (Hendrycks et al., 2020) performance drops by 0.52% due to many low-memory devices (with less than 5.4 GB) being unable to effectively participate in training, which prevents them from contributing their valuable data to the global model. The decline is more significant for LLaMA2-7B, as no device has enough memory to support the fine-tuning process. To pinpoint the root cause, we analyze memory allocation during LLM fine-tuning (Figure 1(c)). The breakdown clearly shows that model parameters are the primary memory consumer, accounting for a dominant share (e.g., 89.6% for LLaMA3-3B), while LoRA modules and activations consume considerably less (0.027% and 10.4%). This finding indicates that existing memory optimization techniques that reduce LoRA parameters or activations are insufficient. Effectively breaking the memory wall requires reducing the memory consumed by the core model parameters.

## 2.3 LAYER PRUNING TO BREAK THE MEMORY WALL

**Pruning Layers for Complexity Reduction.** Motivated by the above observation, we focus on the layer pruning (Men et al., 2024) to reduce model parameters. This technique directly removes entire layers to reduce model complexity, offering inherent flexibility and hardware-agnostic advantages. Additionally, memory savings scale proportionally with the number of pruned layers. We first explore whether layer pruning can decrease model complexity while maintaining model performance. To verify this hypothesis, we conduct experiments with LLaMA2-7B and evaluate the resulting perplexity (PPL) after removing individual layers. Figure 2 shows that significant performance degradation only occurs when removing the initial layers or the last layer, whereas pruning middle

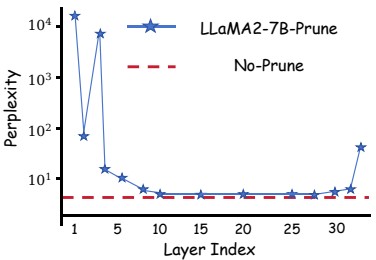

Figure 2: Perplexity tested on the Alpaca-GPT4 dataset (Peng et al., 2023) when pruning one layer.

layers yields minimal performance loss. This finding suggests that *LLMs exhibit significant layer redundancy*, providing an opportunity to implement layer pruning that reduces memory consumption while maintaining model performance.

**Evaluating Multi-Layer Pruning Strategies.** While single-layer pruning yields promising results, the stringent resource constraints of edge devices necessitate more aggressive model compression. We therefore extend our analysis to multi-layer pruning—specifically, pruning ten layers—to further investigate the potential of layer pruning in federated fine-tuning. We evaluate and compare six pruning strategies: 1) *Random*: randomly selecting layers for removal; 2) *Middle*: pruning consecutive layers from the model's middle section; 3) *Norm* (Men et al., 2024): pruning layers with the smallest hidden state norms; 4) *Relative Magnitude (RM)* (Samragh et al.,

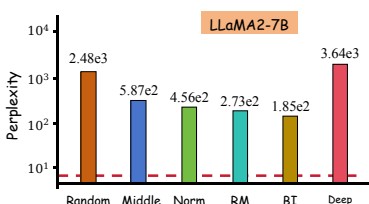

Figure 3: Perplexity tested on the Alpaca-GPT4 dataset (Peng et al., 2023) when pruning ten layers.

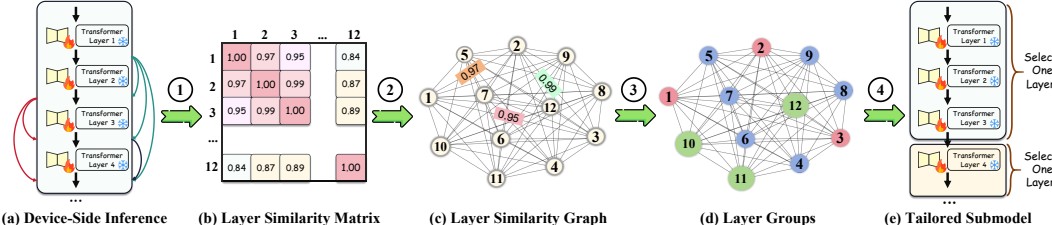

(a) Device-Side Inference    (b) Layer Similarity Matrix    (c) Layer Similarity Graph    (d) Layer Groups    (e) Tailored Submodel

Figure 4: Overview of the functionality-driven layer orchestration mechanism, illustrated with a 12-layer Transformer model, which comprises four key steps: 1) layer similarity computation, 2) graph construction, 3) graph partitioning, and 4) submodel construction.

2023): quantifying layer importance via the metric $||\frac{f(x)}{x+f(x)}||$, where $f$ represents the layer transformation function; 5) *Block Influence (BI)* (Men et al., 2024): measuring layer significance based on the magnitude of feature modifications; 6) *Deep* (Gromov et al., 2024): prioritizing deeper layers removal due to more knowledge retention in shallow layers. Figure 3 shows that when applied to multiple layer pruning scenarios, these strategies result in substantial performance degradation. This deterioration highlights that existing fixed-rule and heuristic-based approaches become inadequate as pruning intensity increases, *underscoring the need for more scientific pruning strategies to preserve model capabilities under aggressive compression.*

## 3 OUR METHOD: FEDPRUNER

Drawing from these empirical findings, we introduce FEDPRUNER, a macro-micro synergistic pruning framework that systematically coordinates cross-device pruning by jointly considering layer functionality (Section 3.1) and layer contribution (Section 3.2). Furthermore, we propose a fine-grained variant (Section 3.3) that independently prunes architectural components, thereby preserving critical structural elements.

### 3.1 MACRO: FUNCTIONALITY-DRIVEN LAYER ORCHESTRATION

The performance degradation in existing layer pruning methods arises from severe functional impairment of the pruned submodels, weakening their ability to capture essential features. This is because neural networks process information hierarchically, progressing from low-level patterns to high-level semantics. Unprincipled pruning disrupts this hierarchy. To address this issue, we propose a macro-level functionality-driven layer orchestration (FDLO) mechanism. FDLO adaptively partitions layers into groups based on their functional characteristics, and then selects *one* representative layer from each group to construct device-specific submodels. The grouping is directly governed by the device's memory budget. For instance, if the budget affords training three layers, the model is divided into three functional groups accordingly.

This design is motivated by the observation that certain layers exhibit highly similar behaviors, naturally forming functional groups. By pruning redundant layers within each group, FDLO preserves the submodel's hierarchical feature extraction capability while satisfying device memory constraints. Figure 4 provides an overview of the FDLO mechanism, which consists of four key steps:

**STEP 1: Layer Similarity Computation.** Given a model with $N$ layers, each device performs inference on its local data and computes inter-layer output similarity using Centered Kernel Alignment (CKA) (Kornblith et al., 2019), as shown in Figure4(a). Specifically, the similarity between outputs $A_i$ and $A_j$ of layers $i$ and $j$ is calculated via Equation 2, where HSIC denotes the Hilbert–Schmidt Independence Criterion (Gretton et al., 2005) (see Appendix A for the formal definition). This yields a layer similarity matrix $\mathbf{W}$, exemplified in Figure 4(b).

$$\text{CKA}(A_i, A_j) = \frac{\text{HSIC}(A_i, A_j)}{\sqrt{\text{HSIC}(A_i, A_i)\,\text{HSIC}(A_j, A_j)}}. \tag{2}$$

**STEP 2: Graph Construction.** Using the similarity matrix $\mathbf{W}$, each device constructs a complete undirected graph $G = (\mathcal{V}, \mathcal{E})$, where the vertex set $\mathcal{V}$ corresponds to network layers and the edge set $\mathcal{E}$ encodes inter-layer similarities, as shown in Figure 4(c).

**STEP 3: Graph Partitioning.** We partition the constructed graph $G = (\mathcal{V}, \mathcal{E})$ into $K$ disjoint groups $\{\mathcal{G}_1, \ldots, \mathcal{G}_K\}$, where $\mathcal{G}_\phi \cap \mathcal{G}_\omega = \emptyset$ for $\phi \neq \omega$ and $\bigcup_{k=1}^{K} \mathcal{G}_k = \mathcal{V}$. The partitioning process consists of three key steps: (i) Computing the Laplacian Matrix: Given the layer similarity matrix $\mathbf{W} \in \mathbb{R}^{N \times N}$, where $w_{i,j}$ represents the pairwise similarity between layers $i$ and $j$, we construct the degree matrix $\mathbf{D} = \mathrm{diag}(d_1, \ldots, d_N)$, where each diagonal element $d_i = \sum_{j=1}^{N} w_{i,j}$ represents the sum of edge weights connected to vertex $i$. The Laplacian matrix is calculated as $\mathbf{L} = \mathbf{D} - \mathbf{W}$.

(ii) Eigendecomposition: We perform eigenvalue decomposition on $\mathbf{L}$ and extract the $K$ smallest non-zero eigenvalues with their corresponding eigenvectors. These eigenvectors form the optimal low-dimensional representation of the graph, where $K$ is determined by the number of layers that the device memory can afford. (iii) Layer Grouping: Using the selected $K$ eigenvectors, we construct a feature matrix $\mathbf{E} \in \mathbb{R}^{N \times K}$ where each row represents a vertex as a $K$-dimensional vector. We then apply k-means to partition these vertices into $K$ groups, as shown in Figure 4(d). The whole procedure can be formulated as Equation 3.

$$\{\mathcal{G}_1, ..., \mathcal{G}_K\} = \text{k-means}(\mathbf{E} \in \mathbb{R}^{N \times K}),$$
$$\text{where } \mathbf{E} = \text{EigVectors}_K \left( \mathrm{diag}(\sum_{j=1}^{N} w_{i,j})_{i=1}^{N} - \mathbf{W} \right). \tag{3}$$

**STEP 4: Submodel Construction.** Following layer grouping results $\{\mathcal{G}_1, ..., \mathcal{G}_K\}$, we select *one* representative layer from each group to construct a $K$-layer submodel that satisfies the device memory constraints while preserving hierarchical feature extraction capabilities, as shown in Figure 4(e).

Additionally, to address the challenge of data heterogeneity, we group layers using locally computed similarity matrices, enabling each device to independently organize layers in a way that best matches its unique data distribution. Furthermore, considering that layer functionality evolves during fine-tuning, we adaptively recalculate the similarity matrices for each device to maintain synchronization between layer organizations and the model's state, thereby enhancing training stability.

### 3.2 MICRO: IMPORTANCE-AWARE LAYER SELECTION

While the macro-level layer orchestration mechanism effectively preserves the submodel's functional completeness, selecting the most representative layer within each group remains non-trivial. This challenge stems from the fact that although layers within the same group share similar functionalities, they still demonstrate notable differences in data processing. These differences become particularly pronounced when the model is divided into fewer groups, as each group inevitably encompasses a broader spectrum of layer characteristics, leading to larger intra-group variations.

To address this challenge, we propose a micro-level importance-aware layer selection (IALS) strategy that assigns retention probabilities to layers based on their contributions to model performance. Specifically, we quantify layer contributions by analyzing input-output feature disparities, where larger disparities indicate greater feature transformations and thus higher importance. By prioritizing these high-impact layers, IALS effectively preserves critical computational paths and accelerates model convergence. The micro-level IALS strategy involves three key steps:

**STEP 1: Layer Importance Quantification.** Given the layer similarity matrix $\mathbf{W} \in \mathbb{R}^{N \times N}$, where $w_{n-1,n}$ measures the input–output feature similarity of layer $n$, we define its importance score as $\sigma_n = 1 - w_{n-1,n}$, capturing the magnitude of feature transformation and the layer's contribution to the model output.

**STEP 2: Selection Probability Generation.** Layer importance scores are converted to selection probabilities via a group-wise softmax. For layer $n$ in $\mathcal{G}_k$, the selection probability is defined as:

$$p_n = \frac{\exp(\sigma_n)}{\sum_{m \in \mathcal{G}_k} \exp(\sigma_m)}. \tag{4}$$

**STEP 3: Representative Layer Selection.** For each group, *one* representative layer is stochastically sampled based on selection probabilities to construct device-specific submodels, while non-selected layers are pruned. This strategy effectively exploits high-importance layers while exploring diverse layer combinations, facilitating efficient discovery of optimal submodel architectures.

Following micro-level layer selection, each device obtains its customized submodel and conducts local fine-tuning. Only the updated LoRA parameters are transmitted to the central server for aggregation, as detailed in Appendix B. The complete workflow of FEDPRUNER, encompassing both macro-level orchestration and micro-level selection, is described in Appendix C. Furthermore, we provide the convergence analysis of FEDPRUNER in Appendix D.

### 3.3 FINE-GRAINED: COMPONENT-LEVEL PRUNING

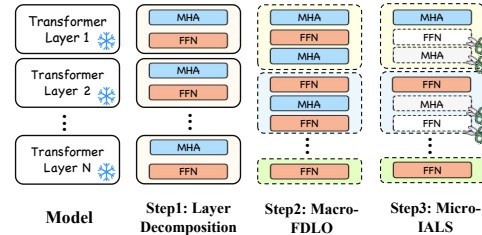

In this section, we introduce FEDPRUNER$^+$, a fine-grained variant of FEDPRUNER that extends the macro–micro synergistic pruning framework from the layer level to the component level. This design builds on two key observations: 1) **Modular Independence**: Each transformer layer consists of two major components, Multi-Head Attention (MHA) and Feed-Forward Network (FFN), which operate independently while maintaining dimensional consistency. This enables selective component pruning without introducing dimensional mismatches. 2)

Figure 5: Workflow of the FEDPRUNER$^+$.

**Heterogeneous Functionality**: MHA and FFN serve distinct roles, with MHA capturing contextual relationships and FFN performing non-linear transformations. These observations motivate us to treat them as independent prunable units.

The workflow of FEDPRUNER$^+$, illustrated in Figure 5, differs from FEDPRUNER in requiring layer decomposition before executing macro-micro synergistic pruning, and proceeds in three steps: 1) **Layer Decomposition**: Each layer is decomposed into its constituent MHA and FFN components, which are treated as independent prunable units. 2) **MACRO-FDLO**: Macro-level layer orchestration is performed on the decomposed architecture, producing $2K$ groups. 3) **MICRO-IALS**: Within each group, micro-level selection identifies *one* representative component for submodel construction. In this way, FEDPRUNER$^+$ achieves finer control over pruning granularity.

## 4 EXPERIMENTS

### 4.1 EXPERIMENTAL SETUP

Consistent with OpenFedLLM (Ye et al., 2024), we evaluate FEDPRUNER and its fine-grained variant on instruction tuning tasks across three LLaMA-based LLMs with varying parameter scales: TinyLLaMA (1.1B) (Zhang et al., 2024b), LLaMA2-7B, and LLaMA2-13B (Touvron et al., 2023). All models are fine-tuned on the Alpaca-GPT4 dataset (Peng et al., 2023). Evaluation is conducted on both close-ended and open-ended benchmarks. The close-ended evaluation suite includes TruthfulQA (Lin et al., 2022) (truthfulness), MMLU (Hendrycks et al., 2020) (knowledge), IFEval (Zhou et al., 2023) (instruction following), and BBH (Suzgun et al., 2022) (reasoning). The open-ended evaluation adopts Vicuna-Bench (Chiang et al., 2023) and MT-Bench (Zheng et al., 2024) to assess multi-turn conversational ability. Additionally, we utilize accuracy as the primary metric for close-ended benchmarks, while employing GPT-4 Scoring to evaluate open-ended tasks. More experimental details are provided in Appendix E.2.

### 4.2 BASELINES

We compare FEDPRUNER with a theoretical baseline (LoRA), which assumes all devices have sufficient memory to support local fine-tuning, and report Zero-Shot performance as a lower bound. We further benchmark against two categories of methods: 1) **Memory-Unaware Methods**: FedIT (Zhang et al., 2024a), DoFIT (Xu et al., 2024), and FedSA-LoRA (Guo et al., 2024); 2) **Memory-Aware Methods**: FLoRA (Wang et al., 2024), FwdLLM (Xu et al., 2023), and FedRA (Su et al., 2024). Detailed baseline descriptions are provided in Appendix F.

Table 1: Performance evaluation on instruction tuning tasks. **Bold** and underlined values denote the best and second-best performance, respectively. The symbol "−" indicates that the method is not applicable due to the lack of devices with sufficient memory resources to support local fine-tuning, thereby degenerating to Zero-Shot performance. The $PR$ represents the device participation rate.

| Method | Close-Ended Benchmark | | | | Average | Open-Ended Benchmark | | | Average | PR |
|---|---|---|---|---|---|---|---|---|---|---|
| | TruthfulQA | MMLU | IFEval | BBH | | Vicuna | MT-1 | MT-2 | | |
| **TinyLLaMA** (Zhang et al., 2024b) | | | | | | | | | | |
| Zero-Shot | 37.58 | 24.72 | 16.05 | 25.91 | 26.07 | 5.52 | 1.61 | 1.18 | 2.77 | / |
| **Memory Unaware** — FedIT | 37.87 | 24.94 | 16.19 | 26.03 | 26.26 | 5.68 | 1.78 | 1.29 | 2.92 | 60% |
| DoFIT | 38.81 | 25.64 | 18.72 | 27.42 | 27.65 | 6.13 | 2.27 | 1.44 | 3.28 | 60% |
| FedSA-LoRA | 38.63 | 25.56 | 19.01 | 27.51 | 27.68 | 6.25 | 2.35 | 1.51 | 3.37 | 60% |
| **Memory Aware** — FLoRA | 38.47 | 25.35 | 17.58 | 27.24 | 27.16 | 5.84 | 1.96 | 1.33 | 3.04 | 60% |
| FwdLLM | 38.55 | 25.49 | 18.64 | 27.35 | 27.51 | 5.98 | 2.09 | 1.36 | 3.14 | 75% |
| FedRA | 38.74 | 25.52 | 19.23 | 27.57 | 27.77 | 6.26 | 2.40 | 1.57 | 3.41 | 100% |
| FEDPRUNER | 39.16 | 26.18 | 19.87 | 28.09 | 28.33 | 6.87 | 2.81 | 1.92 | 3.87 | 100% |
| FEDPRUNER$^+$ | **39.32** | **26.42** | **20.26** | **28.35** | **28.59** | **7.00** | **2.94** | **2.01** | **3.98** | 100% |
| **Theoretical** — LoRA | 38.58 | 25.46 | 18.67 | 27.33 | 27.51 | 6.02 | 2.13 | 1.35 | 3.17 | 100% |
| **LLaMA2-7B (INT4)** (Touvron et al., 2023) | | | | | | | | | | |
| Zero-Shot | 40.98 | 40.20 | 25.46 | 36.95 | 35.90 | 7.00 | 3.04 | 1.13 | 3.72 | / |
| **Memory Unaware** — FedIT | 41.96 | 40.82 | 26.38 | 37.13 | 36.57 | 7.20 | 3.37 | 1.35 | 3.97 | 15% |
| DoFIT | 43.74 | 41.53 | 28.71 | 38.45 | 38.11 | 7.79 | 4.06 | 1.59 | 4.48 | 15% |
| FedSA-LoRA | 45.27 | 41.77 | 29.60 | 38.62 | 38.82 | 8.03 | 4.28 | 1.82 | 4.71 | 15% |
| **Memory Aware** — FLoRA | 43.58 | 41.35 | 28.06 | 37.94 | 37.73 | 7.57 | 3.85 | 1.47 | 4.30 | 15% |
| FwdLLM | 45.83 | 42.20 | 30.57 | 38.97 | 39.39 | 8.16 | 4.71 | 2.05 | 4.97 | 45% |
| FedRA | 47.66 | 42.41 | 31.67 | 39.36 | 40.28 | 8.21 | 5.12 | 2.27 | 5.20 | 100% |
| FEDPRUNER | 49.95 | 43.64 | 33.72 | 40.65 | 41.99 | 8.32 | 5.64 | 2.81 | 5.59 | 100% |
| FEDPRUNER$^+$ | **50.40** | **44.26** | **34.17** | **40.99** | **42.46** | **8.44** | **5.83** | **2.96** | **5.74** | 100% |
| **Theoretical** — LoRA | 47.57 | 42.45 | 31.76 | 39.28 | 40.27 | 8.18 | 4.77 | 1.98 | 4.98 | 100% |
| **LLaMA2-13B (INT4)** (Touvron et al., 2023) | | | | | | | | | | |
| Zero-Shot | 42.83 | 49.65 | 30.35 | 41.03 | 40.97 | 7.27 | 3.65 | 2.07 | 4.33 | / |
| **Memory Unaware** — FedIT | — | — | — | — | — | — | — | — | — | 0% |
| DoFIT | — | — | — | — | — | — | — | — | — | 0% |
| FedSA-LoRA | — | — | — | — | — | — | — | — | — | 0% |
| **Memory Aware** — FLoRA | — | — | — | — | — | — | — | — | — | 0% |
| FwdLLM | — | — | — | — | — | — | — | — | — | 0% |
| FedRA | 51.69 | 54.93 | 39.11 | 45.39 | 47.78 | 8.46 | 5.54 | 3.28 | 5.76 | 100% |
| FEDPRUNER | 56.13 | 58.09 | 45.62 | 48.46 | 52.08 | 8.59 | 5.92 | 4.16 | 6.22 | 100% |
| FEDPRUNER$^+$ | **57.36** | **58.89** | **46.65** | **48.93** | **52.96** | **8.75** | **6.16** | **4.45** | **6.45** | 100% |
| **Theoretical** — LoRA | 52.40 | 55.45 | 40.33 | 46.14 | 48.58 | 8.37 | 5.17 | 3.01 | 5.52 | 100% |

## 4.3 OVERALL EVALUATION

Table 1 presents the experimental results, which demonstrate that both FEDPRUNER and its fine-grained variant consistently outperform baseline methods across all evaluation benchmarks.

1) **Comparison with Memory-Unaware Methods.** As model size grows, device participation rates for these methods drop sharply—from 60% on TinyLLaMA to 15% on LLaMA2-7B, and to 0% on LLaMA2-13B—causing substantial performance loss. For instance, on close-ended benchmarks, FedIT exhibits a 2.07% average performance degradation on TinyLLaMA, which widens to 5.42% on LLaMA2-7B and ultimately reaches 11.11% on LLaMA2-13B compared to FEDPRUNER. DoFIT and FedSA-LoRA display similar patterns of performance deterioration, further underscoring the scalability limitations of memory-unaware approaches.

2) **Comparison with Memory-Aware Methods.** FLoRA optimizes only LoRA modules, yielding limited memory savings and thus showing performance degradation similar to memory-unaware methods. FwdLLM achieves moderate memory optimization by eliminating the need to store intermediate activations, increasing device participation from 60% to 75% (TinyLLaMA) and 15% to 45% (LLaMA2-7B), but still fails on LLaMA2-13B. FedRA enables local fine-tuning under memory constraints but lacks a systematic layer allocation strategy, leading to a 4.3% (close-ended) and 0.46 (open-ended) average performance drop on LLaMA2-13B compared to FEDPRUNER.

3) **Comparison with Theoretical Baseline.** FEDPRUNER consistently outperforms the theoretical baseline (LoRA). For instance, on LLaMA2-13B, it delivers average performance gains of 3.5% on

closed-ended benchmarks and 0.7 on open-ended benchmarks, while reducing memory consumption by 58.5%. These gains arise from FEDPRUNER's intelligent layer pruning strategy, which selectively fine-tunes the most critical layers with minimal disruption to the model's pre-trained knowledge.

4) **Benefits of Fine-Grained Pruning.** FEDPRUNER$^+$ outperforms FEDPRUNER across all evaluation settings, with gains amplifying as model complexity increases. For example, on close-ended benchmarks, the average performance improvement rises from 0.26% for TinyLLaMA to 0.47% for LLaMA2-7B, and 0.88% for LLaMA2-13B. These gains stem from its fine-grained pruning strategy, which more effectively preserves essential architectural components.

### 4.4 UNDERSTANDING THE MACRO-MICRO SYNERGISTIC PRUNING FRAMEWORK

To better understand the macro-micro synergistic pruning framework, we present a detailed analysis focusing on three critical dimensions: model performance, convergence efficiency, and layer selection patterns. Specifically, we experiment with LLaMA2-7B and compare with an ablated variant (denoted as w/o MM) that omits the synergistic pruning framework. As shown in Figure 6(a), FED-PRUNER improves MMLU accuracy by 1.23% and achieves 1.33× faster convergence compared to w/o MM. To elucidate the fundamental

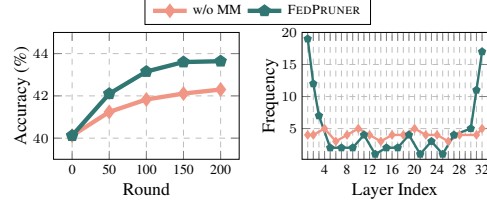

Figure 6: Impact of the macro-micro synergistic pruning framework: (a) MMLU accuracy and (b) layer selection frequency of constructed submodels for a device with 3 GB memory.

reasons driving this performance enhancement, Figure 6(b) illustrates the layer selection patterns during submodel construction for a 3 GB device throughout the federated fine-tuning process. We observe that FEDPRUNER strategically prioritizes head and tail layers in the constructed submodels, diverging from uniform sampling strategy. This scientific submodel construction method effectively improves model performance and training efficiency. Moreover, this observed layer selection pattern aligns with our previous analysis, suggesting a higher degree of redundancy in intermediate layers. More analyses are presented in Appendix G.

### 4.5 UNDERSTANDING THE FINE-GRAINED PRUNING STRATEGY

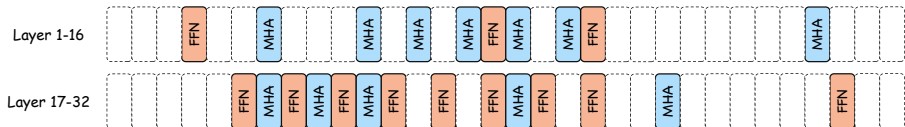

Figure 7: Architecture of the submodel constructed through component-level pruning.

In this section, we investigate the key factors contributing to the success of our fine-grained pruning strategy by analyzing the composition of constructed submodels. Figure 7 illustrates the constructed submodel for a device with 3 GB memory at round 194, with LLaMA2-7B serving as the global model. We observe that component-level pruning enables more flexible architectural configurations. For example, the FFN from layer 2 connects to the MHA from layer 4, indicating that pairing MHA and FFN within the same layer is not always optimal. Notably, the MHA components of layers 4, 6, 7, and 8 are directly connected, suggesting that the conventional alternating MHA–FFN pattern may be suboptimal. FEDPRUNER$^+$ leverages its flexible pruning strategy to enable optimized architectural arrangements, thereby achieving superior performance. Additional experimental results and analysis are presented in Appendix H.

### 4.6 ABLATION STUDY

We further conduct extensive ablation studies to evaluate the contributions of each technique proposed in FEDPRUNER, i.e., functionality-driven layer orchestration (FDLO) and importance-aware layer selection (IALS). The experimental results presented in Table 2 demonstrate that both techniques significantly enhance the model performance, with their benefits becoming more pronounced

Table 2: Ablation study of FEDPRUNER.

| Method | Close-Ended Benchmark | | | | Average |
|---|---|---|---|---|---|
| | TruthfulQA | MMLU | IFEval | BBH | |
| LLaMA2-7B (INT4) | | | | | |
| FEDPRUNER | **49.95** | **43.64** | **33.72** | **40.65** | **41.99** |
| w/o FDLO | 48.14 | 42.93 | 32.44 | 39.58 | 40.77 (-1.22%) |
| w/o IALS | 49.38 | 43.02 | 32.87 | 40.42 | 41.42 (-0.57%) |
| w/o FDLO & IALS | 47.66 | 42.41 | 31.67 | 39.36 | 40.28 (-1.71%) |
| LLaMA2-13B (INT4) | | | | | |
| FEDPRUNER | **56.13** | **58.09** | **45.62** | **48.46** | **52.08** |
| w/o FDLO | 53.24 | 55.76 | 41.47 | 46.45 | 49.23 (-2.85%) |
| w/o IALS | 54.82 | 56.94 | 43.64 | 47.74 | 50.79 (-1.29%) |
| w/o FDLO & IALS | 51.69 | 54.93 | 39.11 | 45.39 | 47.78 (-4.30%) |

Table 3: Overhead analysis of various methods on LLaMA2-7B.

| Method | Resource Consumption (↓) | |
|---|---|---|
| | Time (h) | Communication (GB) |
| FedIT | 3.22 | 6.05 |
| DoFIT | 4.48 | 7.62 |
| FedSA-LoRA | 2.93 | 5.46 |
| FLoRA | 3.61 | 7.46 |
| FwdLLM | 2.98 | 5.58 |
| FedRA | 2.10 | 3.65 |
| LoRA | 2.56 | 5.03 |
| FEDPRUNER | **1.92** | **3.47** |

as model complexity increases. For LLaMA2-7B, removing FDLO and IALS leads to average performance degradation of 1.22% and 0.57%, respectively, while removing both results in a 1.71% decline. The impact is more substantial on LLaMA2-13B, where the performance drops reach 2.85% and 1.29% for individual removals, and 4.3% when both techniques are disabled. These results validate the effectiveness of each technique and the complementary benefits of our proposed macro-micro synergistic pruning framework. Moreover, this empirical evidence further explains that performance degradation in layer importance-based pruning methods primarily stems from their inability to preserve the hierarchical information processing capabilities of the resulting submodels.

## 4.7 OVERHEAD ANALYSIS

Finally, we evaluate FEDPRUNER's efficiency in terms of convergence time and communication cost using LLaMA2-7B as the global model on NVIDIA H800 GPUs. As shown in Table 3, FEDPRUNER outperforms all baselines, achieving up to $2.33\times$ faster convergence and $2.20\times$ lower communication cost. These substantial gains stem from its macro–micro synergistic pruning framework, which systematically coordinates cross-device pruning to accelerate model convergence. Overall, FEDPRUNER is a resource-efficient approach, making it particularly well-suited for federated fine-tuning of LLMs in resource-constrained environments.

## 4.8 PERFORMANCE UNDER DIFFERENT MEMORY CONSTRAINTS

In this section, we evaluate FEDPRUNER's effectiveness under different memory constraints. Specifically, we configure uniform memory constraints across all devices, varying the memory capacity from 25% to 75% of the global model size. For instance, 25% indicates that devices can only accommodate one-fourth of the global model size during local fine-tuning. We benchmark against FedRA as it represents the most closely related work to our approach. Table 4 presents the experimental results, demonstrating that FEDPRUNER consistently outperforms FedRA across all evaluation settings. For LLaMA2-7B, FEDPRUNER achieves average performance improvements of 3.10% at 25% memory budget, 2.54% at 50%, and 2.17% at 75% compared to FedRA.

Table 4: Performance under different memory constraints. The percentage indicates the proportion of global model size each device can accommodate.

| Memory Constraints | Method | Close-Ended Benchmark | | | | Average |
|---|---|---|---|---|---|---|
| | | TruthfulQA | MMLU | IFEval | BBH | |
| LLaMA2-7B (INT4) | | | | | | |
| 25% | FedRA | 42.54 | 41.03 | 27.72 | 37.45 | 37.19 |
| | FEDPRUNER | **46.85** | **42.67** | **31.96** | **39.66** | **40.29 (+3.10%)** |
| 50% | FedRA | 44.93 | 41.62 | 29.57 | 38.34 | 38.62 |
| | FEDPRUNER | **48.67** | **43.16** | **32.74** | **40.05** | **41.16 (+2.54%)** |
| 75% | FedRA | 46.58 | 42.14 | 30.86 | 39.02 | 39.65 |
| | FEDPRUNER | **49.72** | **43.51** | **33.57** | **40.49** | **41.82 (+2.17%)** |
| / | LoRA | 47.57 | 42.45 | 31.76 | 39.28 | 40.27 |
| LLaMA2-13B (INT4) | | | | | | |
| 25% | FedRA | 47.71 | 53.19 | 35.64 | 43.41 | 44.99 |
| | FEDPRUNER | **53.78** | **57.26** | **43.94** | **47.24** | **50.56 (+5.57%)** |
| 50% | FedRA | 50.61 | 54.56 | 37.91 | 44.57 | 46.91 |
| | FEDPRUNER | **55.81** | **58.09** | **45.06** | **47.96** | **51.73 (+4.82%)** |
| 75% | FedRA | 52.53 | 55.42 | 39.58 | 45.50 | 48.26 |
| | FEDPRUNER | **57.04** | **58.67** | **46.15** | **48.59** | **52.61 (+4.35%)** |
| / | LoRA | 52.40 | 55.45 | 40.33 | 46.14 | 48.58 |

The gains are even more pronounced on LLaMA2-13B. Specifically, under a 25% memory constraint on LLaMA2-13B, FEDPRUNER achieves a remarkable 5.57% average accuracy improvement compared to FedRA and even surpasses LoRA by 1.98% while reducing peak memory usage by 75%. These compelling results substantiate the robustness of FEDPRUNER under diverse memory constraints and the effectiveness of its pruning strategy in parameter optimization.

Table 5: Performance evaluation on contemporary SOTA LLMs.

| Method | Close-Ended Benchmark ↑ | | | | |
|--------|------------|------|--------|-----|---------|
| | TruthfulQA | MMLU | IFEval | BBH | Average |
| **LLaMA3.2-3B** (Dubey et al., 2024) | | | | | |
| LoRA | 44.25 | 54.19 | 53.60 | 46.84 | 49.72 |
| FEDPRUNER | 49.62 | 60.43 | 66.58 | 56.71 | **58.34 (+8.62)** |
| **LLaMA3.1-8B** (Dubey et al., 2024) | | | | | |
| LoRA | 48.07 | 63.31 | 47.32 | 62.69 | 55.35 |
| FEDPRUNER | 53.98 | 67.95 | 61.40 | 70.51 | **63.46 (+8.11)** |
| **Qwen2.5-7B** (Yang et al., 2025) | | | | | |
| LoRA | 48.02 | 41.89 | 33.68 | 42.63 | 41.56 |
| FEDPRUNER | 54.65 | 46.70 | 39.51 | 45.96 | **46.71 (+5.15)** |

## 4.9 SCALABILITY WITH STATE-OF-THE-ART LLMS

To rigorously assess the robustness and scalability of FEDPRUNER, we conduct extensive validation on a spectrum of contemporary SOTA models, specifically LLaMA3.2-3B (Dubey et al., 2024), LLaMA3.1-8B (Dubey et al., 2024), and Qwen2.5-7B (Yang et al., 2025). As evidenced in Table 5, FEDPRUNER consistently outperforms the theoretical baseline (LoRA) across all tested architectures. Notably, it achieves average performance gains of 8.62% on LLaMA3.2-3B, 8.11% on LLaMA3.1-8B, and 5.15% on Qwen2.5-7B. These substantial improvements affirm the method's adaptability to diverse model families and scales, solidifying the practical relevance of FEDPRUNER for parameter-efficient federated fine-tuning in real-world scenarios.

## 5 RELATED WORK

**Layer Pruning.** Layer pruning (Gromov et al., 2024) has emerged as a prominent technique for LLM compression by eliminating redundant layers. ShortGPT (Men et al., 2024) reveals that straightforward layer pruning can achieve comparable performance to sophisticated width pruning methods. LaCo (Men et al., 2024) proposes a concise layer-wise structured pruning method that consolidates posterior layers into their preceding counterparts for model compression. Different from existing works that focus on obtaining a compact model, FEDPRUNER employs layer pruning to address the memory constraints in federated fine-tuning.

**Memory-Efficient Federated Fine-Tuning.** Existing memory-efficient federated fine-tuning methods mainly focus on reducing LoRA parameters or intermediate activations. For instance, FLoRA (Wang et al., 2024), FlexLoRA (Bai et al., 2024), and HETLoRA (Cho et al., 2024) adaptively assign LoRA ranks based on device resource constraints. Building on this, LEGENDS (Liu et al., 2025), Fed-pilot (Zhang et al., 2024c), and Fed-HeLLo (Zhang et al., 2025) advance this direction by refining layer-wise LoRA allocation to simultaneously enhance model performance and resource efficiency. Alternatively, FedKSeed (Qin et al., 2023) and FwdLLM (Xu et al., 2023) employ zeroth-order optimization to bypass the storage of activations. However, these approaches fail to address the memory constraints as they still require retaining the full model in memory. In contrast, FEDPRUNER addresses this bottleneck by strategically pruning layers to reduce model parameters, thereby fundamentally mitigating memory constraints.

## 6 CONCLUSION

In this paper, we introduce FEDPRUNER, a novel federated fine-tuning paradigm that employs layer pruning to overcome the memory constraints of participating devices. To systematically coordinate the pruning process across devices, we develop a macro-micro synergistic pruning framework. Furthermore, we propose a fine-grained component-level pruning framework to precisely preserve essential components. Extensive experiments on benchmark datasets demonstrate that both FEDPRUNER and its fine-grained variant consistently outperform state-of-the-art baseline methods.

## REPRODUCIBILITY STATEMENT

In the supplementary materials, we provide an anonymous code package that includes source files, experimental configurations, and algorithm implementations to ensure reproducibility of the results. Practical deployment and runtime settings are documented in Section 4.1 and Appendix E.2. The complete algorithmic workflow is detailed in Appendix C, while the formal assumptions and full derivations of the convergence analysis appear in Appendix D. The evaluation benchmarks are specified in Appendix E.1, and descriptions of all baseline methods are provided in Appendix F.

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

## A   HILBERT-SCHMIDT INDEPENDENCE CRITERION (HSIC)

The HSIC between two random variables $X$ and $Z$ is computed as:

$$
\begin{aligned}
\text{HSIC}(X, Z) = \mathbb{E}_{X,Z,X',Z'}\left[k_X(X, X')k_{Z'}(Z, Z')\right] + \mathbb{E}_{X,X'}\left[k_X(X, X')\right]\mathbb{E}_{Z'}\left[k_Z(Z, Z')\right] \\
- 2\mathbb{E}_{X,Z}\left[\mathbb{E}_{X'}\left[k_X(X, X')\right]\mathbb{E}_{Z'}\left[k_Z(Z, Z')\right]\right],
\end{aligned}
\tag{5}
$$

where $k_X$ and $k_Z$ are kernel functions operating on independently drawn pairs $(X, X')$ and $(Z, Z')$, respectively, and $\mathbb{E}_{XZ}$ denotes the expectation over $X$ and $Z$.

## B   AGGREGATION STRATEGY IN FEDPRUNER

We denote the LoRA parameters for layer $n$ at communication round $r$ as $\theta_{\text{LoRA}}^{n,r} = (\mathbf{A}^{n,r}, \mathbf{B}^{n,r})$. Let $\mathcal{S}$ represent the complete set of selected devices participating in the federated fine-tuning process, while $\mathcal{S}_n \subseteq \mathcal{S}$ indicates the specific subset of devices designated for updating the parameters of the $n^{\text{th}}$ layer. The aggregation mechanism for the LoRA parameters of layer $n$ at the subsequent round $r + 1$ can be elegantly formulated as:

$$
(\mathbf{A}^{n,r+1}, \mathbf{B}^{n,r+1}) = \begin{cases} \left(\lambda_1\mathbf{A}_1^{n,r} \oplus \lambda_2\mathbf{A}_2^{n,r} \oplus \cdots \oplus \lambda_{|S_n|}\mathbf{A}_{|S_n|}^{n,r}, \right. \\ \left. \quad \mathbf{B}_1^{n,r} \oplus \mathbf{B}_2^{n,r} \oplus \cdots \oplus \mathbf{B}_{|S_n|}^{n,r}\right), & \text{if } \mathcal{S}_n \neq \emptyset \\ (\mathbf{A}^{n,r}, \mathbf{B}^{n,r}), & \text{if } \mathcal{S}_n = \emptyset \end{cases}
\tag{6}
$$

In this formulation, $\lambda_s$ represents the weighting coefficient proportional to each device's data contribution, defined as $\lambda_s = \frac{|D_s|}{\sum_{s=1}^{|S_n|}|D_s|}$, where $|D_s|$ quantifies the cardinality of device $s$'s local dataset. The operator "$\oplus$" denotes the dimension-specific concatenation procedure (Wang et al., 2024): applying vertical concatenation for matrix $\mathbf{A}$ and horizontal concatenation for matrix $\mathbf{B}$.

When $\mathcal{S}_n = \emptyset$, indicating that no participating devices have been assigned to update the $n^{\text{th}}$ layer's LoRA parameters, these parameters persist unchanged from the previous round. Conversely, when $\mathcal{S}_n$ contains at least one device, the server aggregates the local parameter updates from all devices in this subset to derive the updated LoRA parameters for the $n^{\text{th}}$ layer.

Here is a concrete example to illustrate the aggregation process. Consider a global model with six layers, $\Theta = \{\theta_1, ..., \theta_6\}$. In round $r$, two devices are selected:

- Device 1 updates the layers $\{\theta_1, \theta_3, \theta_5\}$.
- Device 2 updates the layers $\{\theta_2, \theta_3, \theta_5\}$.

The overall aggregation process at the server is performed as follows:

- **Overlapping Layers** $(\theta_3, \theta_5)$: For layers updated by both selected devices, the server computes a weighted aggregation of the updates from both devices.
- **Exclusive Layers** $(\theta_1, \theta_2)$: For layers updated by a single device, the update is applied directly.
- **Unselected Layers** $(\theta_4, \theta_6)$: Layers not selected by any device remain unchanged.

**Algorithm 1** The workflow of FEDPRUNER

---

**Require:** Global model $\Theta$, $R$ rounds
**Ensure:** Optimized global model $\Theta^*$
 1: **for** $r = 0$ **to** $R - 1$ **do**
 2:   **if** $r == 0$ **then**
 3:     Distribute global model $\Theta$ to all devices
 4:   **else**
 5:     Distribute all LoRA parameters to selected device set $\mathcal{S}$
 6:   **end if**
 7:   **for** each device $s \in \mathcal{S}$ **do**
 8:     Recompute the local similarity matrix using a batch of data
 9:     Execute macro-level FDLO (Section 3.1)
10:     Execute micro-level IALS (Section 3.2)
11:     Perform the local fine-tuning process
12:     Upload only updated LoRA parameters
13:   **end for**
14:   Server performs parameter aggregation using Equation 6
15: **end for**

---

## C  THE ALGORITHM OF FEDPRUNER

The pseudocode of FEDPRUNER is summarized in Algorithm 1.

## D  CONVERGENCE ANALYSIS OF FEDPRUNER

In this section, we provide a theoretical convergence analysis for FEDPRUNER. Our analysis builds upon established results from federated optimization under partial model updates (Wang et al., 2022; Wu et al., 2025a) and layer pruning (Sun et al., 2024). We denote: 1) $\Theta^r = \{\theta^{1,r}, \theta^{2,r}, \ldots, \theta^{N,r}\}$ as the global model parameters (including LoRA parameters) at round $r$. 2) $\mathcal{S}$ as the set of devices selected for local updates in each round. 3) $\mathcal{S}_n$ as the subset of devices that specifically update the $n$-th layer at round $r$.

Let $f_s(\Theta)$ be the local objective function for device $s$, and let $F(\Theta)$ be the global objective function defined as the average loss over all devices:

$$F(\Theta) = \frac{1}{|\mathcal{S}|} \sum_{s \in \mathcal{S}} f_s(\Theta). \tag{7}$$

To capture partial model updates, we define the submodel $\Theta_s$ for device $s$ as the set of layers selected via FDLO (Section 3.1) and IALS (Section 3.2). Correspondingly, each device $s$ updates only the sub-parameters $\Theta_s$ during local fine-tuning, leaving the other parameters unchanged.

### D.1  PRELIMINARIES AND KEY ASSUMPTIONS

Similar to standard federated learning convergence analyses (Li et al., 2019; 2022; 2023) (e.g., FedAvg), we make the following assumptions:

**Assumption D.1 (Smoothness)** *Each local objective function $f_s(\Theta)$ satisfies the L-smoothness property, which can be formally expressed as: for any parameter vectors $\Theta_1, \Theta_2$,*

$$\|\nabla f_s(\Theta_1) - \nabla f_s(\Theta_2)\| \leq L\|\Theta_1 - \Theta_2\|, \tag{8}$$

*where $L > 0$ is the smoothness constant that bounds the Lipschitz continuity of the gradient.*

**Assumption D.2 (Unbiased Gradient and Bounded Variance)** *For each device $s$, the stochastic gradient $\nabla f_s(\Theta; \xi)$ computed on a randomly sampled mini-batch $\xi$ satisfies:*

$$\mathbb{E}[\nabla f_s(\Theta; \xi)] = \nabla f_s(\Theta), \tag{9}$$

*establishing the unbiasedness of the gradient estimator, and*

$$\mathbb{E}\big[\|\nabla f_s(\Theta;\xi) - \nabla f_s(\Theta)\|^2\big] \leq \sigma^2, \tag{10}$$

*where $\sigma^2$ is a finite positive constant that uniformly bounds the variance of the stochastic gradient across all devices and parameter values.*

**Assumption D.3 (Partial Layer Update)** *At communication round $r$, each participating device $s \in \mathcal{S}$ updates only a device-specific subset of layers corresponding to its assigned submodel $\Theta_s$. Formally, if $\Theta$ represents the complete model parameters, the global update can be expressed as:*

$$\Theta^{r+1} = \Theta^r + \Delta^r, \quad where \; \Delta^r \in \mathbb{R}^{|\Theta^r|}, \tag{11}$$

*with the following property: for any parameter $\theta_n \in \Theta$ that is not part of device $s$'s submodel assignment, the corresponding element in $\Delta^r$ contributed by device $s$ is zero. Only parameters within the assigned submodel $\Theta_s$ may receive non-zero updates from device $s$.*

These assumptions align with established practices in federated optimization literature and provide the necessary foundation for our convergence analysis, particularly when dealing with heterogeneous submodel assignments and partial parameter updates across the network.

## D.2 ONE-ROUND ANALYSIS

We begin by analyzing the expected decrease in the global loss $F(\Theta)$ after one communication round, denoted by going from $\Theta^r$ to $\Theta^{r+1}$. According to Assumption D.1, we have:

$$F(\Theta^{r+1}) \leq F(\Theta^r) + \langle \nabla F(\Theta^r), \Theta^{r+1} - \Theta^r \rangle + \frac{L}{2}\|\Theta^{r+1} - \Theta^r\|^2. \tag{12}$$

We use the aggregated gradient update from Equation 6, where each device $s$ updates only the submodel $\Theta_s$. Let $\eta$ be the learning rate. Then the update on each device $s$ for the layer $n \in \Theta_s$ is approximately:

$$\Delta_n^{s,r} = -\eta \nabla_{\Theta_n} f_s(\Theta^r), \tag{13}$$

and $\Delta_n^{s,r} = \mathbf{0}$ if layer $n$ is not selected by device $s$. Thus, the server aggregates updates by:

$$\Delta^r = \sum_{s \in \mathcal{S}} \sum_{n \in \Theta_s} \lambda_{s,n} \Delta_n^{s,r}, \tag{14}$$

where $\lambda_{s,n}$ are the weighting coefficients for aggregation (see Equation 6). Substituting 14 into 12, and taking expectation, we get:

$$\mathbb{E}\big[F(\Theta^{r+1})\big] \leq \mathbb{E}\big[F(\Theta^r)\big] - \eta \, \mathbb{E}\Big[\langle \nabla F(\Theta^r), \sum_{s \in \mathcal{S}} \sum_{n \in \Theta_s} \lambda_{s,n} \nabla_{\Theta_n} f_s(\Theta^r) \rangle\Big]$$
$$+ \frac{L\eta^2}{2}\mathbb{E}\Big[\big\|\sum_{s \in \mathcal{S}} \sum_{n \in \Theta_s} \lambda_{s,n} \nabla_{\Theta_n} f_s(\Theta^r)\big\|^2\Big]. \tag{15}$$

By the unbiased gradient assumption (Assumption D.2) and the fact that $F(\Theta)$ is the average over $f_s(\Theta)$, we can relate $\nabla F(\Theta^r)$ to $\nabla f_s(\Theta^r)$. Further, because only some devices and layers are selected, the update reflects a partial average of the gradient. Consequently, standard bounding techniques yield:

$$\mathbb{E}\big[F(\Theta^{r+1})\big] \leq \mathbb{E}\big[F(\Theta^r)\big] - \frac{\eta}{|\mathcal{S}|} \sum_{s \in \mathcal{S}} \sum_{n \in \Theta_s} \lambda_{s,n} \, \mathbb{E}\Big[\|\nabla_{\Theta_n} f_s(\Theta^r)\|^2\Big]$$
$$+ \frac{L\eta^2}{2}\mathbb{E}\Big[\big\|\sum_{s \in \mathcal{S}} \sum_{n \in \Theta_s} \lambda_{s,n} \nabla_{\Theta_n} f_s(\Theta^r)\big\|^2\Big] + \text{(variance term)}. \tag{16}$$

Under bounded variance (Assumption D.2) and standard norms bounding, the variance term can be controlled by a constant factor of $\sigma^2$. This leads to an upper bound for the variance-induced deviation in the full gradient.

### D.3 MULTI-ROUND CONVERGENCE

Summing over $r = 0, \ldots, R - 1$ and rearranging terms, we obtain a standard telescoping series argument:

$$\sum_{r=0}^{R-1} \left( \mathbb{E}[F(\Theta^{r+1})] - \mathbb{E}[F(\Theta^r)] \right) \leq \sum_{r=0}^{R-1} \left( -\alpha \, \mathbb{E}[\|\nabla F(\Theta^r)\|^2] + \beta \right), \tag{17}$$

where $\alpha$ and $\beta$ are constants dependent on $\eta$, $L$, $\sigma^2$, and the weighting coefficients $\{\lambda_{s,n}\}$. The left-hand side telescopes to $\mathbb{E}[F(\Theta^R)] - \mathbb{E}[F(\Theta^0)]$. Thus, we get:

$$\mathbb{E}[F(\Theta^R)] - \mathbb{E}[F(\Theta^0)] \leq -\alpha \sum_{r=0}^{R-1} \mathbb{E}[\|\nabla F(\Theta^r)\|^2] + R\,\beta. \tag{18}$$

Rearranging and dividing by $R$, we have:

$$\frac{1}{R} \sum_{r=0}^{R-1} \mathbb{E}[\|\nabla F(\Theta^r)\|^2] \leq \frac{\mathbb{E}[F(\Theta^0)] - \mathbb{E}[F(\Theta^R)]}{\alpha\, R} + \frac{\beta}{\alpha}. \tag{19}$$

Since $F(\Theta)$ is bounded below (common in deep learning with non-negative loss functions) we have that $\mathbb{E}[F(\Theta^0)] - \mathbb{E}[F(\Theta^R)]$ is finite. As $R \to \infty$, the term $\frac{\mathbb{E}[F(\Theta^0)] - \mathbb{E}[F(\Theta^R)]}{\alpha\, R}$ vanishes, and hence the average squared gradient norm converges to a bounded value dependent on $\frac{\beta}{\alpha}$.

Since $\beta$ can be made arbitrarily small by choosing sufficiently small learning rate $\eta$ or utilizing FDLO and IALS to refine the partial layer selection probabilities to mitigate variance, $\nabla F(\Theta^r)$ converges in expectation to $\mathbf{0}$. This implies that FEDPRUNER converges to a stationary point of $F(\Theta)$.

### D.4 IMPACT OF THE MACRO-MICRO SYNERGISTIC PRUNING ON CONVERGENCE

FEDPRUNER performs pruning at both macro (functionality-driven layer orchestration) and micro (importance-aware layer selection) levels. The key impact on convergence involves the reduced parameter space and potential variance in gradient estimates due to partial updates. However, these effects are incorporated in the weighting coefficients $\{\lambda_{s,n}\}$ and the variance bound $\sigma^2$. Specifically:

- **FDLO:** Although each device may prune layers differently based on correlated layer functionalities, the overall update $\Delta^r$ remains an unbiased approximation of the full gradient (extended to the submodel). Aggregation across all participating devices ensures that the regularly updated parameters converge with the sufficiently expressive submodels.

- **IALS:** Representative layer selection within each group introduces another variance component in gradient updates. Nonetheless, as shown in Equation 14, these updates remain unbiased with respect to the submodel's gradient. As the number of communication rounds grows, the exploration of different representative layers allows devices to approximate the true submodel gradient adequately.

Therefore, under standard smoothness and bounded variance conditions, FEDPRUNER achieves convergence to a stationary point of the global objective $F(\Theta)$. This convergence is guaranteed by unbiased local gradients, careful aggregation via Equation 6, and the dynamic recalculation of model pruning in FDLO and IALS.

### D.5 CONCLUSION

In summary, we have shown that FEDPRUNER converges to a stationary point under common assumptions in federated optimization. The macro-level functionality-driven layer orchestration (FDLO) and micro-level importance-aware layer selection (IALS) do not invalidate the unbiasedness of local gradient updates. Consequently, our theoretical analysis confirms that the partial update scheme maintains convergence guarantees akin to standard federated learning approaches.

# E ADDITIONAL EXPERIMENTAL SETUP

## E.1 DATASETS

The evaluation datasets are described as follows:

- TruthfulQA (Lin et al., 2022) is a benchmark designed to evaluate the truthfulness of language models' responses. It consists of 817 questions across 38 categories, with a focus on common misconceptions and false beliefs. The questions are specifically crafted to test whether models can avoid generating false or misleading information, often stemming from patterns learned during training.

- MMLU (Massive Multitask Language Understanding) (Hendrycks et al., 2020) is a benchmark that assesses models' knowledge across 57 subjects, including science, humanities, mathematics, and professional fields. It contains 15,908 multiple-choice questions designed to test both broad knowledge and in-depth understanding, sourced from a variety of educational materials, including practice exams for standardized tests and professional certifications.

- IFEval (Instruction Following Evaluation) (Zhou et al., 2023) evaluates models' ability to follow verifiable instructions that can be objectively assessed for compliance. It includes 25 distinct instructions with multiple variants and 541 prompts, designed to test various aspects of instruction adherence such as task comprehension, constraint satisfaction, and format compliance.

- BBH (BIG-Bench Hard) (Suzgun et al., 2022) is a benchmark focused on challenging reasoning tasks that require advanced cognitive abilities. It consists of 23 tasks specifically chosen for their difficulty, evaluating skills such as logical deduction, mathematical reasoning, and abstract thinking.

- Vicuna-Bench (Chiang et al., 2023) is a comprehensive evaluation framework designed to assess the conversational abilities of language models through a variety of dialogue scenarios. It emphasizes natural language interactions and evaluates key aspects such as coherence, relevance, contextual understanding, and response quality.

- MT-Bench (Zheng et al., 2024) is a benchmark designed to evaluate the multi-turn dialogue capabilities of language models, featuring complex conversation scenarios that require maintaining context across multiple exchanges. It assesses the model's ability to sustain coherence, consistency, and relevance throughout extended interactions, capturing human-like conversational preferences.

## E.2 IMPLEMENTATION DETAILS

Following the setup of OpenFedLLM (Ye et al., 2024), we partition the Alpaca-GPT4 dataset across 20 devices. During each training round, 10% of the devices are randomly selected to perform local training, where each device executes 10 local update steps with a batch size of 16. For optimization, we utilize the AdamW optimizer alongside a cosine learning rate schedule, where the learning rate gradually decreases from 1e-4 to 5e-6 over time. The maximum input sequence length is set to 512 tokens (Ye et al., 2024). When computing the similarity matrix, we utilize only a single batch of data for the calculation, thereby minimizing computational overhead. In FEDPRUNER$^+$, LoRA is integrated into both MHA and FFN.

To emulate practical deployment conditions, we randomly assign 3–9 GB of available memory to participating devices, guided by memory profiling results from a variety of mobile hardware (Xu et al., 2023; Tam et al., 2024). If a device does not have enough memory to afford the local fine-tuning process, it is excluded from participating. Additionally, to support efficient deployment of LLaMA2 series models on resource-limited edge devices, we perform INT4 quantization for model compression. We conduct 150 rounds of federated fine-tuning for TinyLLaMA, 200 rounds for LLaMA2-7B, and 300 rounds for LLaMA2-13B. All experiments are repeated multiple times and the reported results are averaged across runs to ensure statistical reliability.

Table 6: Layer grouping results, intra-group layer selection probabilities, and selected layers for sub-model construction at round 6. Note that the selected layers displayed represent a single realization sampled based on the computed probabilities.

| Group Index | The Layer Index within the Group | Layer Selection Probability within the Group | Final Selected Layer |
|---|---|---|---|
| $\mathcal{G}_1$ | [1] | [1.0] | layer 1 |
| $\mathcal{G}_2$ | [2] | [1.0] | layer 2 |
| $\mathcal{G}_3$ | [3] | [1.0] | layer 3 |
| $\mathcal{G}_4$ | [4] | [1.0] | layer 4 |
| $\mathcal{G}_5$ | [5] | [1.0] | layer 5 |
| $\mathcal{G}_6$ | [6,7,8,9,10,11,12,13,14,15,16, 17,18,19,20,21,22,23,24,25,26] | [0.05, 0.05, 0.05, 0.05, 0.05, 0.05, 0.05, 0.05, 0.04, 0.04, 0.03, 0.04, 0.05, 0.05, 0.05, 0.05, 0.05, 0.05, 0.05, 0.05, 0.05] | layer 13 |
| $\mathcal{G}_7$ | [27] | [1.0] | layer 27 |
| $\mathcal{G}_8$ | [28] | [1.0] | layer 28 |
| $\mathcal{G}_9$ | [29] | [1.0] | layer 29 |
| $\mathcal{G}_{10}$ | [30] | [1.0] | layer 30 |
| $\mathcal{G}_{11}$ | [31] | [1.0] | layer 31 |
| $\mathcal{G}_{12}$ | [32] | [1.0] | layer 32 |

## F  BASELINE DESCRIPTION

**Memory-Unaware Methods:**

- 1) FedIT (Zhang et al., 2024a): This method integrates LoRA with FedAvg to perform instruction tuning.
- 2) DoFIT (Xu et al., 2024): A domain-aware approach mitigates catastrophic forgetting by utilizing tailored initialization and aggregation strategies for LoRA weights.
- 3) FedSA-LoRA (Guo et al., 2024): This method uploads only the $\mathbf{A}$ matrices, which encode generalizable knowledge, to the server, while keeping the device-specific $\mathbf{B}$ matrices on local devices to preserve personalization.

**Memory-Aware Methods:**

- 4) FLoRA (Wang et al., 2024): This method assigns different ranks to devices based on their available resources and proposes a stacking-based strategy to aggregate heterogeneous LoRA modules.
- 5) FwdLLM (Xu et al., 2023): This method employs zeroth-order optimization to update LoRA parameters, removing the requirement to store intermediate activations.
- 6) FedRA (Su et al., 2024): This approach randomly generates layer allocation matrices for devices to create submodels that fit within their memory constraints, making it the most closely related work to ours.

**Theoretical Baseline:**

- 7) LoRA: This baseline assumes that all devices have sufficient memory to support local fine-tuning, representing the theoretical upper bound.

## G  UNDERSTANDING THE MACRO-MICRO SYNERGISTIC PRUNING FRAMEWORK

For the device with 3 GB available memory, it can accommodate the fine-tuning process of a sub-model with 12 transformer layers for LLaMA2-7B (INT4). To provide deeper insights into the macro-micro synergistic pruning framework, we present the layer similarity matrices, layer grouping results, and the selection probabilities of layers within each group at two critical stages: the early phase (round 6) and the late phase (round 194) of federated fine-tuning. The grouping results and

Table 7: Layer grouping results, intra-group layer selection probabilities, and selected layers for submodel construction at round 194. Note that the selected layers displayed represent a single realization sampled based on the computed probabilities.

| Group Index | The Layer Index within the Group | Layer Selection Probability within the Group | Final Selected Layer |
|---|---|---|---|
| $\mathcal{G}_1$ | [1,2,3,4,5] | [0.16, 0.18, 0.21, 0.23, 0.22] | layer 4 |
| $\mathcal{G}_2$ | [6,7] | [0.55, 0.45] | layer 6 |
| $\mathcal{G}_3$ | [8] | [1.0] | layer 8 |
| $\mathcal{G}_4$ | [9] | [1.0] | layer 9 |
| $\mathcal{G}_5$ | [10] | [1.0] | layer 10 |
| $\mathcal{G}_6$ | [11,12,13,14,15,16,17,18,19] | [0.11, 0.11, 0.11, 0.11, 0.12, 0.11, 0.11, 0.11, 0.11] | layer 16 |
| $\mathcal{G}_7$ | [20] | [1.0] | layer 20 |
| $\mathcal{G}_8$ | [21] | [1.0] | layer 21 |
| $\mathcal{G}_9$ | [22] | [1.0] | layer 22 |
| $\mathcal{G}_{10}$ | [23,24] | [0.48,0.52] | layer 24 |
| $\mathcal{G}_{11}$ | [25,26] | [0.46,0.54] | layer 25 |
| $\mathcal{G}_{12}$ | [27,28,29,30,31,32] | [0.16, 0.18, 0.20, 0.17, 0.15, 0.14] | layer 31 |

layer selection probabilities are shown in Table 6 and Table 7, respectively. At round 6, we observe stronger similarities among middle layers, with $\mathcal{G}_6$ containing 21 layers. This indicates the presence of numerous functionally similar layers in the model, validating the effectiveness of our macro-level functionality-driven layer orchestration mechanism and providing opportunities for layer pruning. As training progresses, this clustering effect gradually extends to both ends of the model, as evidenced at round 194 where $\mathcal{G}_1$ clusters layers 1-5, $\mathcal{G}_6$ clusters layers 11-19, and $\mathcal{G}_{12}$ clusters layers 27-32. This demonstrates the importance of dynamic layer grouping. Additionally, we observe significant variations in layer selection probabilities within $\mathcal{G}_1$ and $\mathcal{G}_{12}$, validating the effectiveness of our micro-level importance-aware layer selection strategy. Figure 8 presents the corresponding similarity matrices, further illustrating this layer clustering phenomenon.

## H  UNDERSTANDING THE FINE-GRAINED PRUNING STRATEGY

We present the layer grouping results, intra-group layer selection probabilities, and selected layers for submodel construction obtained through component-level pruning for a 3 GB device at rounds 6 and 194. Compared to layer-level pruning which divides the model into 12 groups, component-level pruning partitions the model into 24 groups, where one component is selected from each group for submodel construction. The experimental results are shown in Tables 8 and 9, respectively. Unlike layer-level pruning, we observe that component-level pruning enables more fine-grained functional grouping of the model, where MHA and FFN from the same transformer layer can be assigned to different groups, while offering greater flexibility in constructing submodels.

## I  BROADER IMPACTS

**Positive Impacts.** FEDPRUNER significantly reduces memory resources required for local LLM fine-tuning through intelligent layer pruning, enabling more resource-constrained devices to participate in the federated fine-tuning process, effectively utilizing their data to cultivate higher-performing models. This capability is particularly valuable in healthcare, environmental monitoring, and education sectors, especially in underserved regions where resources are scarce but the potential societal benefits of AI are substantial.

Beyond accessibility, FEDPRUNER promotes environmental sustainability in AI by minimizing communication overhead and computational requirements, directly reducing energy consumption and carbon emissions. This framework further strengthens privacy protection by keeping more computation on users' devices, reducing the need to centralize sensitive data and aligning with modern data protection principles.

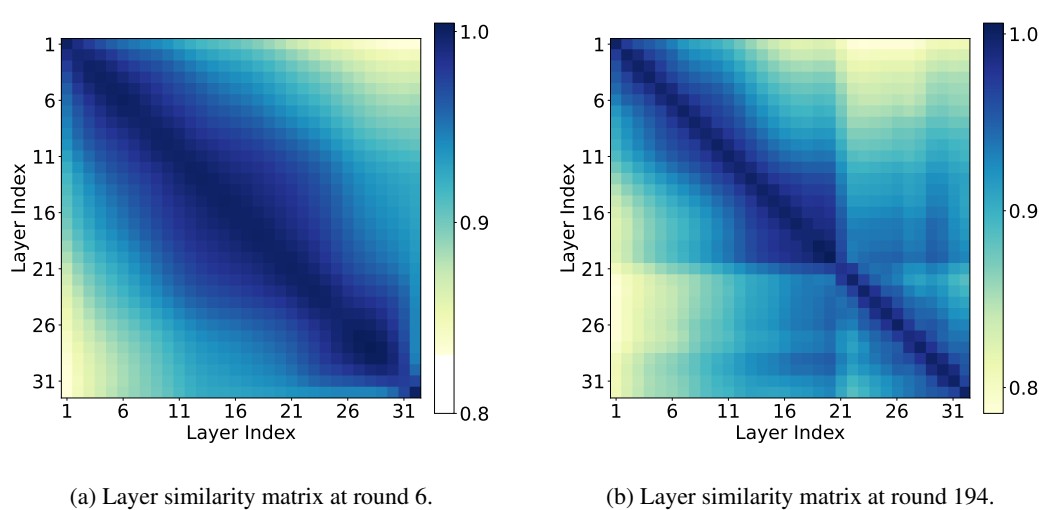

(a) Layer similarity matrix at round 6.    (b) Layer similarity matrix at round 194.

Figure 8: Layer similarity matrices at different rounds for the device with 3 GB available memory.

Table 8: Fine-grained layer grouping results, intra-group layer selection probabilities, and selected layers for submodel construction at round 6, where $n^M$ and $n^N$ denote the MHA and FFN components of the $n$-th layer, respectively. The red highlights indicate novel assembly patterns emerging in submodel construction. Note that the selected layers displayed represent a single realization sampled based on the computed probabilities.

| Group Index | The Layer Index within the Group | Layer Selection Probability within the Group | Final Selected Layer |
|---|---|---|---|
| $\mathcal{G}_1$ | $[1^M]$ | $[1.0]$ | $1^M$ |
| $\mathcal{G}_2$ | $[1^N]$ | $[1.0]$ | $1^N$ |
| $\mathcal{G}_3$ | $[2^M]$ | $[1.0]$ | $2^M$ |
| $\mathcal{G}_4$ | $[2^N]$ | $[1.0]$ | $2^N$ |
| $\mathcal{G}_5$ | $[3^M]$ | $[1.0]$ | $3^M$ |
| $\mathcal{G}_6$ | $[3^N]$ | $[1.0]$ | $3^N$ |
| $\mathcal{G}_7$ | $[4^M]$ | $[1.0]$ | $4^M$ |
| $\mathcal{G}_8$ | $[4^N]$ | $[1.0]$ | $4^N$ |
| $\mathcal{G}_9$ | $[5^M]$ | $[1.0]$ | $5^M$ |
| $\mathcal{G}_{10}$ | $[5^N]$ | $[1.0]$ | $5^N$ |
| $\mathcal{G}_{11}$ | $[6^M]$ | $[1.0]$ | $6^M$ |
| $\mathcal{G}_{12}$ | $[6^N,7^M,7^N,8^M,8^N,9^M,9^N,10^M,10^N,11^M,11^N,12^M,12^N,$ $13^M,13^N,14^M,14^N,15^M,15^N,16^M,16^N,17^M,17^N,$ $18^M,18^N,19^M,19^N,20^M,20^N,21^M,21^N,22^M,22^N,$ $23^M,23^N,24^M,24^N,25^M,25^N,26^M,26^N]$ | $[0.025, 0.025, 0.025, 0.025, 0.025, 0.025, 0.025, 0.025, 0.025, 0.025, 0.025,$ $0.025, 0.025, 0.025, 0.025, 0.025, 0.024, 0.023, 0.023, 0.022, 0.022,$ $0.021, 0.022, 0.022, 0.023, 0.023, 0.025, 0.025, 0.025, 0.025, 0.025,$ $0.025, 0.025, 0.025, 0.025, 0.025, 0.025, 0.025, 0.025, 0.025, 0.025]$ | $16^N$ |
| $\mathcal{G}_{13}$ | $[27^M]$ | $[1.0]$ | $27^M$ |
| $\mathcal{G}_{14}$ | $[27^N]$ | $[1.0]$ | $27^N$ |
| $\mathcal{G}_{15}$ | $[28^M]$ | $[1.0]$ | $28^M$ |
| $\mathcal{G}_{16}$ | $[28^N]$ | $[1.0]$ | $28^N$ |
| $\mathcal{G}_{17}$ | $[29^M]$ | $[1.0]$ | $29^M$ |
| $\mathcal{G}_{18}$ | $[29^N]$ | $[1.0]$ | $29^N$ |
| $\mathcal{G}_{19}$ | $[30^M]$ | $[1.0]$ | $30^M$ |
| $\mathcal{G}_{20}$ | $[30^N]$ | $[1.0]$ | $30^N$ |
| $\mathcal{G}_{21}$ | $[31^M]$ | $[1.0]$ | $31^M$ |
| $\mathcal{G}_{22}$ | $[31^N]$ | $[1.0]$ | $31^N$ |
| $\mathcal{G}_{23}$ | $[32^M]$ | $[1.0]$ | $32^M$ |
| $\mathcal{G}_{24}$ | $[32^N]$ | $[1.0]$ | $32^N$ |

**Negative Impacts.** As a general-purpose framework for memory-efficient federated fine-tuning, FEDPRUNER does not introduce negative societal impacts beyond those inherent to the underlying machine learning systems it optimizes. The technique is application-agnostic and designed to improve efficiency rather than enable new capabilities that could be misused. However, we acknowledge that any technology making AI systems more accessible and deployable could potentially amplify existing risks if applied to systems designed for harmful purposes. By lowering the resource

Table 9: Fine-grained layer grouping results, intra-group layer selection probabilities, and selected layers for submodel construction at round 194, where $n^M$ and $n^N$ denote the MHA and FFN components of the $n$-th layer, respectively. The red highlights indicate novel assembly patterns emerging in submodel construction. Note that the selected layers displayed represent a single realization sampled based on the computed probabilities.

| Group Index | The Layer Index within the Group | Layer Selection Probability within the Group | Final Selected Layer |
|---|---|---|---|
| $\mathcal{G}_1$ | $[1^M, 1^N, 2^M, 2^N, 3^M]$ | [0.16, 0.18, 0.21, 0.23, 0.22] | $2^N$ |
| $\mathcal{G}_2$ | $[3^N, 4^M, 4^N, 5^M, 5^N]$ | [0.23, 0.22, 0.16, 0.18, 0.21] | $4^M$ |
| $\mathcal{G}_3$ | $[6^M, 6^N]$ | [0.55, 0.45] | $6^M$ |
| $\mathcal{G}_4$ | $[7^M, 7^N]$ | [0.52, 0.48] | $7^M$ |
| $\mathcal{G}_5$ | $[8^M]$ | [1.0] | $8^M$ |
| $\mathcal{G}_6$ | $[8^N]$ | [1.0] | $8^N$ |
| $\mathcal{G}_7$ | $[9^M, 9^N]$ | [0.53, 0.47] | $9^M$ |
| $\mathcal{G}_8$ | $[10^M]$ | [1.0] | $10^M$ |
| $\mathcal{G}_9$ | $[10^N]$ | [1.0] | $10^N$ |
| $\mathcal{G}_{10}$ | $[11^M, 11^N, 12^M, 12^N, 13^M, 13^N, 14^M, 14^N, 15^M, 15^N, 16^M, 16^N, 17^M, 17^N, 18^M, 18^N, 19^M]$ | [0.058, 0.058, 0.058, 0.058, 0.058, 0.059, 0.060, 0.060, 0.062, 0.060, 0.060, 0.059, 0.058, 0.058, 0.058, 0.058, 0.058] | $15^M$ |
| $\mathcal{G}_{11}$ | $[19^N]$ | [1.0] | $19^N$ |
| $\mathcal{G}_{12}$ | $[20^M]$ | [1.0] | $20^M$ |
| $\mathcal{G}_{13}$ | $[20^N]$ | [1.0] | $20^N$ |
| $\mathcal{G}_{14}$ | $[21^M]$ | [1.0] | $21^M$ |
| $\mathcal{G}_{15}$ | $[21^N]$ | [1.0] | $21^N$ |
| $\mathcal{G}_{16}$ | $[22^M]$ | [1.0] | $22^M$ |
| $\mathcal{G}_{17}$ | $[22^N]$ | [1.0] | $22^N$ |
| $\mathcal{G}_{18}$ | $[23^M, 23^N]$ | [0.51, 0.49] | $23^N$ |
| $\mathcal{G}_{19}$ | $[24^M, 24^N]$ | [0.46, 0.54] | $24^N$ |
| $\mathcal{G}_{20}$ | $[25^M]$ | [1.0] | $25^M$ |
| $\mathcal{G}_{21}$ | $[25^N]$ | [1.0] | $25^N$ |
| $\mathcal{G}_{22}$ | $[26^M, 26^N]$ | [0.43, 0.57] | $26^N$ |
| $\mathcal{G}_{23}$ | $[27^M, 27^N, 28^M, 28^N, 29^M, 29^N]$ | [0.15, 0.18, 0.20, 0.17, 0.16, 0.14] | $28^M$ |
| $\mathcal{G}_{24}$ | $[30^M, 30^N, 31^M, 31^N, 32^M, 32^N]$ | [0.16, 0.17, 0.20, 0.18, 0.15, 0.14] | $31^N$ |

barrier for fine-tuning language models, FEDPRUNER could inadvertently facilitate the deployment of AI systems without adequate safeguards in place.

We encourage practitioners to apply FEDPRUNER within established ethical guidelines for AI development and deployment, with appropriate consideration for fairness, accountability, and transparency. Implementers should conduct thorough impact assessments before deploying systems optimized with our framework, especially in sensitive domains or applications affecting vulnerable populations.

# J MORE EXPERIMENTS

## J.1 COMPARISON WITH STATE-OF-THE-ART PRUNING METHODS

To further substantiate the robustness and efficacy of FEDPRUNER, we extend our evaluation on TinyLLaMA to include three advanced pruning baselines: DepthFL (Kim et al., 2022), FedMef (Huang et al., 2024), and ShortGPT (Men et al., 2024). As evidenced in Table 10, FEDPRUNER consistently surpasses all competing methods, delivering average performance gains of up to 2.22%. This consistent superiority stems from FEDPRUNER's distinct design, which effectively mitigates key limitations inherent in these baselines:

- Versus DepthFL: DepthFL creates submodels by depth, which often leads to deep layers being trained only by a small fraction of high-memory devices. This constraint leads to undertraining of critical deep layers. In contrast, FEDPRUNER utilizes dynamic orchestration to ensure balanced and comprehensive training across all layers.
- Versus FedMef: FedMef is optimized for CNNs, where intermediate activations are considered the dominant bottleneck. This assumption is inaccurate for modern LLMs, where the model parameters constitute the primary bottleneck (as shown in Figure 1(c)). Furthermore, FedMef does not sufficiently account for data heterogeneity.

Table 10: Performance comparison against advanced pruning methods on TinyLLaMA.

| Method | Close-Ended Benchmark ↑ | | | | |
|--------|-------------|------|--------|-----|---------|
| | TruthfulQA | MMLU | IFEval | BBH | Average |
| DepthFL | 37.72 | 24.85 | 16.14 | 26.00 | 26.18 (-2.15) |
| FedMef | 37.68 | 24.79 | 16.05 | 25.93 | 26.11 (-2.22) |
| ShortGPT | 37.91 | 25.00 | 16.18 | 26.10 | 26.30 (-2.03) |
| FEDPRUNER | 39.16 | 26.18 | 19.87 | 28.09 | **28.33** |

Table 11: Performance comparison with advanced LoRA optimization methods on LLaMA2-7B.

| Method | Close-Ended Benchmark ↑ | | | | |
|--------|-------------|------|--------|-----|---------|
| | TruthfulQA | MMLU | IFEval | BBH | Average |
| Fed-pilot | 44.15 | 41.83 | 29.14 | 38.85 | 38.49 (-3.50) |
| Fed-HeLLo | 43.86 | 41.62 | 28.71 | 38.43 | 38.16 (-3.83) |
| FlexLoRA | 43.67 | 41.46 | 28.17 | 38.06 | 37.84 (-4.15) |
| HETLoRA | 43.42 | 41.21 | 27.93 | 37.76 | 37.58 (-4.41) |
| FEDPRUNER | 49.95 | 43.64 | 33.72 | 40.65 | 41.99 |

- Versus ShortGPT: ShortGPT relies on pruning layers based on simple block influence scores. This method inherently overlooks the complex functional specificity and collaborative dependencies among different layers. Our approach, the macro-micro synergistic pruning framework, is explicitly designed to model and leverage these dependencies for more effective and less destructive pruning.

## J.2 COMPARISON WITH STATE-OF-THE-ART LoRA OPTIMIZATION METHODS

Fed-pilot (Zhang et al., 2024c), Fed-HeLLo (Zhang et al., 2025), FlexLoRA (Bai et al., 2024), and HETLoRA (Cho et al., 2024) represent significant advancements in memory-efficient federated fine-tuning, primarily focusing on optimizing LoRA modules (e.g., via rank adaptation or layer-wise allocation). However, as illustrated in Figure 1(c), LoRA constitutes a negligible fraction of the total memory footprint (e.g., merely 0.018% for LLaMA2-7B). Consequently, optimizing these modules yields minimal savings and fails to effectively alleviate the prohibitive memory constraints imposed by LLMs.

To empirically substantiate this, we conduct comparative experiments on LLaMA2-7B. As detailed in Table 11, FEDPRUNER achieves average performance gains of up to 4.41% over these baselines. This performance advantage is driven by enhanced client inclusivity: whereas competing approaches are compelled to exclude low-memory devices due to insufficient memory reduction, FEDPRUNER enables comprehensive participation. Importantly, we position FEDPRUNER as orthogonal and complementary to these existing works. We envision a synergistic pipeline that first employs FEDPRUNER to derive a memory-efficient submodel, followed by applying methods like Fed-HeLLo to optimize LoRA allocation within that compact structure.

## J.3 COMPARISON WITH MORE RELATED WORK

To further substantiate the superiority of FEDPRUNER, we benchmark it against additional related works—including FedMef (Huang et al., 2024), FlexLoRA (Bai et al., 2024), and LEGENDS (Liu et al., 2025)—on TinyLLaMA. The results in Table 12 confirm the consistent advantage of FEDPRUNER, which achieves average improvements of up to 2.22%. This performance gap stems from fundamental design distinctions: FedMef is tailored for CNNs and fails to account for the fact that model parameters constitute the primary bottleneck in LLMs; meanwhile, FlexLoRA and LEGENDS focus on optimizing LoRA modules, yielding only marginal memory savings. Consequently, these baselines are unable to accommodate resource-constrained devices, inevitably degrad-

Table 12: Performance comparison with other related works on TinyLLaMA.

| Method | Close-Ended Benchmark ↑ | | | | |
|---|---|---|---|---|---|
| | TruthfulQA | MMLU | IFEval | BBH | Average |
| FedMef | 37.68 | 24.79 | 16.05 | 25.93 | 26.11 (-2.22) |
| FlexLoRA | 38.52 | 25.41 | 17.62 | 27.26 | 27.20 (-1.13) |
| LEGENDS | 38.58 | 25.53 | 17.81 | 27.42 | 27.34 (-0.99) |
| FEDPRUNER | 39.16 | 26.18 | 19.87 | 28.09 | **28.33** |

Table 13: Performance evaluation with varying number of clients.

| Method | Close-Ended Benchmark ↑ | | | | |
|---|---|---|---|---|---|
| | TruthfulQA | MMLU | IFEval | BBH | Average |
| **Number of Clients = 10** | | | | | |
| FedIT | 38.42 | 25.38 | 17.65 | 26.59 | 27.01 |
| FEDPRUNER | 39.71 | 26.53 | 21.44 | 28.75 | **29.11 (+2.10)** |
| **Number of Clients = 20** | | | | | |
| FedIT | 37.87 | 24.94 | 16.19 | 26.03 | 26.26 |
| FEDPRUNER | 39.16 | 26.18 | 19.87 | 28.09 | **28.33 (+2.07)** |
| **Number of Clients = 30** | | | | | |
| FedIT | 37.78 | 24.85 | 16.15 | 26.00 | 26.20 |
| FEDPRUNER | 39.05 | 26.11 | 19.76 | 28.02 | **28.24 (+2.04)** |
| **Number of Clients = 50** | | | | | |
| FedIT | 37.74 | 24.76 | 16.08 | 25.94 | 26.13 |
| FEDPRUNER | 39.10 | 26.09 | 19.61 | 27.95 | **28.19 (+2.06)** |

ing global model performance by excluding valuable local data. In contrast, FEDPRUNER enables full (100%) device participation, translating directly into significant performance gains.

### J.4 SCALABILITY WITH VARYING NUMBER OF CLIENTS

To assess the scalability of FEDPRUNER, we conduct additional experiments on TinyLLaMA by varying the total number of participating clients across the set $\{10, 20, 30, 50\}$. As presented in Table 13, FEDPRUNER demonstrates robust performance across different population sizes, consistently maintaining a clear advantage over FedIT. Notably, even with a larger client population of 50, FEDPRUNER achieves an average accuracy improvement of 2.06% over the baseline, thereby effectively validating the scalability of our approach.

### J.5 ROBUSTNESS TO CLIENT PARTICIPATION RATIOS

We further investigate the impact of varying client participation rates by testing ratios of $\{10\%, 20\%, 50\%\}$ on TinyLLaMA. As reported in Table 14, FEDPRUNER demonstrates strong robustness, consistently maintaining high accuracy and outperforming the baseline across all settings. Notably, FEDPRUNER achieves an average performance improvement of 2.07% even under a low participation rate of 10%. Furthermore, we observe a clear positive correlation: as the participation rate increases, the model performance improves correspondingly.

Table 14: Performance evaluation with varying participation ratios.

| Method | Close-Ended Benchmark ↑ | | | | |
|---|---|---|---|---|---|
| | TruthfulQA | MMLU | IFEval | BBH | Average |
| **Participation Ratio = 10%** | | | | | |
| FedIT | 37.87 | 24.94 | 16.19 | 26.03 | 26.26 |
| FEDPRUNER | 39.16 | 26.18 | 19.87 | 28.09 | **28.33 (+2.07)** |
| **Participation Ratio = 20%** | | | | | |
| FedIT | 38.45 | 25.50 | 17.72 | 26.60 | 27.07 |
| FEDPRUNER | 39.72 | 26.71 | 21.58 | 28.79 | **29.20 (+2.13)** |
| **Participation Ratio = 50%** | | | | | |
| FedIT | 38.63 | 25.54 | 18.13 | 26.88 | 27.30 |
| FEDPRUNER | 39.90 | 27.04 | 22.06 | 28.96 | **29.49 (+2.19)** |

## K    DISCUSSION

**Distinction from General Pruning Strategies.** Fundamentally, FEDPRUNER differs from general pruning methods (Ling et al., 2024; Men et al., 2024), which typically generate a single, static compressed model with a fixed target size for all scenarios. In contrast, FEDPRUNER is a dynamic framework specifically engineered to address the dual challenges of heterogeneity in FL. First, regarding resource heterogeneity, FEDPRUNER moves beyond fixed-size pruning by enabling client-specific model construction, where each client's individual memory budget strictly constrains the submodel structure (e.g., layer numbers). Second, to mitigate data heterogeneity, our approach is intrinsically data-driven. In contrast to data-agnostic methods, FEDPRUNER leverages local distributions to quantify inter-layer similarities. This informs the FDLO process, ensuring the submodel is specialized for the client's specific data characteristics. This per-client, memory-aware, and data-driven adaptation constitutes the core advantage of FEDPRUNER over general-purpose pruning.

**Additional Computational Overhead.** The overhead introduced by FDLO (Steps 1-3) is negligible compared to the standard fine-tuning process. Specifically, the cost of Step 1 is dominated by a one-time forward pass on a single batch of data. This initialization cost is computationally insignificant relative to the extensive local training. Quantitatively, for LLaMA2-7B (batch size 16), this single pass incurs approximately 54.3 TFLOPs, whereas a single local fine-tuning round consumes ~2232.0 TFLOPs. Thus, Step 1 constitutes a mere 2.4% of the total local computation. Regarding Steps 2 and 3, operations are performed on a compact $N \times N$ similarity matrix (where $N$ is the layer count) rather than the high-dimensional parameter space. For LLaMA2-7B ($N = 32$), this involves processing a tiny $32 \times 32$ matrix. The execution time for these steps is on the order of milliseconds, rendering them virtually cost-free for edge devices.

**Comparison with Traditional Heterogeneous FL Methods.** While HeteroFL (Diao et al., 2020), AdaptiveFL (Jia et al., 2024), and DepthFL (Kim et al., 2022) are established baselines for resource heterogeneity, they are fundamentally unsuitable for the federated fine-tuning of LLMs for two primary reasons. 1) Paradigm Mismatch: These methods are tailored for training smaller models (e.g., CNNs) from scratch. This objective diverges significantly from our focus on fine-tuning pre-trained LLMs, which entails distinct challenges related to model scale and parameter preservation. 2) Architectural Incompatibility: Many techniques, such as the channel pruning used in HeteroFL and AdaptiveFL, are specific to CNN structures and cannot be directly applied to Transformer attention blocks. Among them, DepthFL is the most relevant as it employs depth-based pruning. However, its heuristic submodel allocation is ill-suited for LLMs, causing severe training imbalance: critical deep layers are updated only by a few high-capacity clients, leading to undertraining. Empirical results in Table 10 validate this flaw: DepthFL suffers an average performance drop of 2.15% on TinyL-LaMA compared to FEDPRUNER. This gap widens on larger models, with a significant degradation of 5.21% on LLaMA2-7B, confirming its inability to be effectively applied to federated fine-tuning.

## L    IMPLEMENTATION DETAILS OF LAYER SIMILARITY CALCULATION

To ensure the feasibility of Step 1 (Layer Similarity Computation) on resource-constrained devices, we employ a memory-efficient **pipelined, layer-by-layer inference workflow**. This mechanism guarantees that operations remain strictly within the device's memory budget through a three-step process:

1. **Adaptive Chunk Loading:** The device loads only the initial subset of layers (e.g., layers 1-3) that fits within its current available memory.

2. **Compute, Cache, and Release:** A forward pass is performed on this chunk using a single data batch. Once the necessary output activations are cached and stored, the device immediately offloads the layer parameters to free memory.

3. **Sequential Iteration:** The device then loads the subsequent chunk (e.g., layers 4-6), utilizing the cached activations from the previous block as input. This cycle repeats until the forward pass is complete for all layers.

This workflow ensures that the peak memory footprint during CKA calculation never exceeds the device's limit, allowing the process to adaptively scale to specific hardware constraints.

## M    IMPLEMENTATION DETAILS OF DETERMINING $K$

The $K$ is determined via a rigorous analytical model grounded in memory profiling, ensuring scalability and generalization across heterogeneous devices. Unlike empirical trial-and-error approaches, our quantitative determination proceeds as follows:

- **Profile Base Cost ($M_{\text{base}}$):** We first quantify the fixed memory overhead required to initialize the model context. This encompasses non-prunable components (e.g., token embeddings, output heads).

- **Profile Layer Cost ($M_{\text{layer}}$):** We then profile the peak memory consumption for fine-tuning a single Transformer block. This metric encapsulates the combined footprint of parameters, intermediate activations, and optimizer states.

- **Profile System Cost ($M_{\text{system}}$):** We also quantify the device-specific system memory overhead, such as the CUDA context footprint, upon deploying the model to the target hardware.

- **Calculate $K$:** Given a client device with a memory budget $M_{\text{device}}$, the maximum viable layer count $K$ is derived analytically:

$$K = \left\lfloor \frac{M_{\text{device}} - M_{\text{base}} - M_{\text{system}}}{M_{\text{layer}}} \right\rfloor . \tag{20}$$

This calculated $K$ serves as the constraint for our macro-level FDLO graph partitioning. Crucially, this analytical approach is model- and device-agnostic, relying solely on standard profiling data to adapt universally to diverse architectures (e.g., LLaMA, Qwen) and edge capacities. When system memory usage is disregarded, $K$ is calculated by:

$$K = \left\lfloor \frac{M_{\text{device}} - M_{\text{base}}}{M_{\text{layer}}} \right\rfloor . \tag{21}$$

## N    FUTURE WORK

A promising avenue for future research lies in generalizing FEDPRUNER to multimodal and generative frontiers. Given that FEDPRUNER is driven by layer-wise utility dynamics (via FDLO and IALS) rather than task-specific heuristics, it holds theoretical transferability to a broader spectrum of architectures. We aim to specifically investigate multimodal adaptation by extending FEDPRUNER to MLLMs (e.g., Qwen-VL (Bai et al., 2025), LLaVA (Liu et al., 2023)), examining how layer-wise pruning interacts with cross-modal feature alignment and the distinct gradient dynamics of visual

versus textual encoders. Concurrently, we plan to validate the framework on generative diffusion backbones (Borji, 2022), verifying whether the block-wise utility observed in LLMs translates effectively to the iterative denoising process required for high-fidelity generation. Exploration in these domains is pivotal for democratizing the training of versatile foundation models within memory-constrained FL ecosystems.

## O    THE USE OF LARGE LANGUAGE MODELS

For transparency, we confined the use of ChatGPT to language refinement tasks only. Its contribution was restricted to light editing of author-written text, such as grammar correction, wording adjustments, and improvements in readability. The model was not involved in generating research ideas, designing methods, writing code, analyzing data, preparing figures or tables, or drafting technical content. All scientific content and results were solely developed and validated by the authors. Any suggestions from the tool were carefully reviewed and, when incorporated, further refined to avoid plagiarism, inaccuracies, or fabricated content. The authors assume full responsibility for the manuscript; the LLM is not listed as an author.

