# OpenReview forum: "Prune to Fit: Enabling Federated Fine-Tuning within Edge Memory Budgets"
_ICLR.cc/2026/Conference — Submitted to ICLR 2026_

### Official Review · Reviewer_nDop · 2025-10-25

**Soundness:** 3
**Presentation:** 2
**Contribution:** 3
**Rating:** 6
**Confidence:** 3

**Summary:**

The paper proposes FedPruner, an innovative federated fine-tuning paradigm that addresses the memory constraints of edge devices via intelligent layer pruning. Specifically, it employs a macro-micro synergistic pruning framework that jointly considers layer functionality and layer contribution to coordinate the pruning process. At the macro level, the functionality-driven layer orchestration (FDLO) mechanism adaptively partitions layers into groups based on their functional characteristics using Centered Kernel Alignment (CKA) and graph partitioning. At the micro level, the importance-aware layer selection (IALS) strategy selects one representative layer from each group according to its contribution to model performance. Furthermore, the paper introduces 〖FedPruner〗^+, a fine-grained variant of FedPruner that extends the macro–micro synergistic pruning framework from the layer level to the component level.

**Strengths:**

1. The method is technically sound and supported by comprehensive experiments across multiple model scales and benchmarks.

2. The paper provides the theoretical analysis regarding the convergence of the proposed method.

3. The paper is generally well-motivated and easy to follow.

4. The proposed method improves upon existing solutions and achieves notable performance and efficiency enhancements.

**Weaknesses:**

1. The paper states that K is determined by the number of layers that the device memory can afford, but it does not provide a clear quantitative rule or algorithmic procedure for this determination. This lack of specification may hinder the method’s scalability and generalizability when applied to other model architectures or edge devices with varying memory constraints.

2. The tables (e.g., Table 1) do not explicitly specify the evaluation metrics used for each benchmark. This lack of clarification may cause confusion for readers.

3. The workflow (Algorithm 1) does not clearly specify whether the server distributes LoRA parameters corresponding to all layers of the global model or only to the layers involved in each device’s submodel during every communication round.

**Questions:**

1. How is the number of layer groups (K) determined in practice for devices with different memory capacities? Is it empirically obtained by testing different configurations on real devices, or is it estimated analytically based on parameter size and memory constraints?

2. In Table 5, for the sixth group G_6, many layers show a layer selection probability of 0.05. Could the authors clarify why layer 13 was ultimately selected, despite multiple layers having the same probabilities?

3. In Algorithm 1, it appears that during each communication round r, each device recomputes the local similarity matrix using a batch of data. Since this matrix is computed based on the inference results of the global model, could the authors clarify whether the server distributes the LoRA parameters corresponding to all layers of the global model to each device in every round, or only those corresponding to the layers included in each device’s submodel?

---

> ### Author Response · Authors · 2025-11-21
> **Response to Reviewer nDop (1 / 1)**
>
> We sincerely thank you for your insightful and constructive review. We have carefully considered each point you raised and have incorporated corresponding revisions and clarifications into the uploaded revised manuscript. Below are our point-by-point responses to your concerns.
>
>
>
> > **W1 & Q1: Quantitative Determination of $K$.**
>
> We apologize that the specific procedure for determining $K$ was not explicitly detailed in the initial submission. We wish to clarify that $K$ is determined via a rigorous analytical model based on memory profiling, rather than by empirical trial-and-error.
>
> The quantitative determination process is as follows:
>
> 1. **Profile Base Cost ($M_{\text{base}}$):** We first quantify the fixed memory overhead required to initialize the model context. This encompasses non-prunable components (e.g., token embeddings, output heads).
>
> 2. **Profile Layer Cost ($M_{\text{layer}}$):** We then profile the peak memory consumption for fine-tuning a single Transformer block. This  encapsulates the combined footprint of parameters, intermediate activations, and optimizer states.
>
> 3. **Profile System Cost ($M_{\text{system}}$):** We also quantify the device-specific system memory overhead, such as the CUDA context footprint, upon deploying the model to the target hardware.
>
> 4. **Calculate $K$**: Given a client device with a memory budget $M_{\text{device}}$, the maximum viable layer count $K$ is derived analytically:
>
>    $$K = \left\lfloor \frac{M_{\text{device}} - M_{\text{base}}-M_{\text{system}}}{M_{\text{layer}}} \right\rfloor.$$
>
> This calculated $K$ serves as the constraint for macro-level FDLO. This quantitative approach ensures that the method is scalable and generalizable across diverse devices and model architectures (e.g., LLaMA, Qwen) . When system memory usage is disregarded, $K$ is calculated by: $K = \left\lfloor \frac{M_{\text{device}}-M_{\text{base}}}{M_{\text{layer}}} \right\rfloor.$ We have included a detailed description of this determination process in **Appendix M**.
>
>
>
> > **W2: Clarification of Evaluation Metrics.**
>
> Sorry for the confusion. The reported results strictly follow the standard evaluation protocols for these benchmarks.
>
> - **Close-ended Benchmarks:** We use **Accuracy** as the evaluation metric.
> - **Open-ended Benchmarks:** We employ **GPT-4 Scoring** as the evaluation metric.
>
> We have incorporated descriptions of these metrics into **Section 4.1**.
>
>
>
> > **W3 & Q3: Clarification on LoRA Parameter Distribution.**
>
> Sorry for the confusion. We wish to clarify that the server distributes the LoRA parameters for all layers to each device per round.
>
> **Necessity for FDLO:** To perform FDLO, the client must compute the complete layer similarity matrix. This requires the device to perform inference on all layers. After the matrix is computed and FDLO/IALS have selected the $K$ representative layers, the device prunes all other layers and performs fine-tuning on its locally constructed submodel.
>
> **Communication Efficiency:** While this involves downloading all LoRA parameters, the method remains highly communication-efficient. The client only uploads the updated parameters for its $K$-layer submodel, which is a fraction of the total. As shown in Table 3, FedPruner still reduces the total communication cost by up to 2.2x compared to baselines.
>
> We have revised **Algorithm 1** to clarify this process.
>
>
>
> > **Q2: Clarification on Layer Selection.**
>
> We thank the reviewer for this keen observation. It highlights a fundamental feature of our micro-level IALS strategy.
>
> **1.Identical Probabilities:** In Group $\mathcal{G}_{6}$, the middle layers exhibit high functional similarity (corroborated by Fig. 8a). Therefore, our algorithm assigns them nearly identical importance scores, yielding a near-uniform probability distribution (e.g., 0.05).
>
> **2.Stochastic Selection:** The selection of Layer 13 is the result of the stochastic sampling inherent to IALS (Section 3.2, Step 3). The algorithm does not strictly select the layer with the highest probability (argmax).
>
> - **Why Layer 13?** It was selected purely by chance from the distribution. In a subsequent round, Layer 14 or 16 could be selected given the same probabilities.
> - **Design Rationale:** This stochasticity is a deliberate design choice. When layers are functionally similar, it prevents the model from deterministically picking the same layer every time. Instead, it encourages the **exploration of diverse layer combinations**, preventing the model from getting trapped in local optima.
>
> We have added an explicit note to the captions of the corresponding tables **(Tables 6, 7, 8, and 9)** to clarify that the 'Final Selected Layer' represents a snapshot of a single stochastic realization sampled from the displayed probabilities.
>
> ---
>
>
>
> Again, we thank the reviewer for the valuable feedback. Please let us know if there are any other questions or suggestions.

---

### Official Review · Reviewer_Lk6t · 2025-10-28

**Soundness:** 2
**Presentation:** 2
**Contribution:** 2
**Rating:** 4
**Confidence:** 5

**Summary:**

This paper proposes FedPruner, a novel federated fine-tuning framework that addresses device memory constraints via coordinated layer pruning. By combining macro-micro pruning strategies and fine-grained component-level control, the method achieves strong performance across benchmark datasets while significantly reducing memory usage.

**Strengths:**

The paper addresses a highly relevant and practical challenge in federated fine-tuning of large language models.

**Weaknesses:**

1. The statement “fine-tuning LLaMA2-7B (Touvron et al., 2023) demands up to 26.9 GB of memory” may not be entirely accurate. Memory usage depends heavily on input token length and batch size, which can significantly increase GPU memory consumption beyond the reported value.

2. In the introduction, the example states: “compared to optimizing LoRA modules and activations, reducing model parameters is more promising to lower training memory usage.” As mentioned, batch size and input token length values can greatly affect memory distribution, as highlighted in Fed-Pilot [1]. Additionally, the memory costs associated with storing gradients and context should also be taken into account.

3. Miss a problem formulation for memory consumption.

4. Several important related works are missing from the literature review. Fed-Pilot [1] provides a comprehensive formulation of memory consumption during fine-tuning and analyzes both static and dynamic activation memory. Fed-HeLLo [2] explores heuristic and Fisher Information-based strategies for layer-wise LoRA allocation. FlexLoRA [3] and HETLoRA [4] are also relevant baselines that should be included in comparative studies.

5. More tasks and metrics should be evaluated in the experiments shown in Figure 2. Different tasks may rely on different levels of model representations. Since perplexity is only suitable for language modeling or summarization tasks, it may not reflect performance well in downstream tasks like question answering, where accuracy or F1-score would be more appropriate.

6. Memory-efficient methods such as Fed-HeLLo, Fed-pilot, FLoRA, HETLoRA, and FlexLoRA are designed to support both homogeneous and heterogeneous resource settings. Can the proposed method also handle such heterogeneity? If so, how does it aggregate updates from clients with heterogeneous model configurations?

References:

[1] Fed-Pilot: Optimizing LoRA Allocation for Efficient Federated Fine-Tuning with Heterogeneous Clients. arXiv, 2024.

[2] Fed-HeLLo: Efficient Federated Foundation Model Fine-Tuning with Heterogeneous LoRA Allocation. IEEE Transactions on Neural Networks and Learning Systems (TNNLS), 2025.

[3] Federated Fine-Tuning of Large Language Models under Heterogeneous Tasks and Client Resources. NeurIPS, 2024.

[4] Heterogeneous LoRA for Federated Fine-Tuning of On-Device Foundation Models. EMNLP, 2024.

**Questions:**

See Weaknesses.

---

> ### Author Response · Authors · 2025-11-21
> **Response to Reviewer Lk6t (1 / 3)**
>
> We sincerely thank you for your insightful and constructive review. We have carefully considered each point you raised and have incorporated corresponding revisions and clarifications into the uploaded revised manuscript. Below are our point-by-point responses to your concerns.
>
>
>
> > **W1: Memory Profiling Details and Context.**
>
> We appreciate the opportunity to clarify the experimental settings behind this measurement. The reported 26.9 GB memory for LLaMA2-7B was profiled under the following specific settings: unquantized model parameters (FP32/FP16), a batch size of 16, and a maximum sequence length of 512. Under these settings, the memory breakdown is as follows:
>
> - **Model Parameters:** ~25,587 MB
> - **Activations:** ~1,985 MB
> - **Optimizer States:** ~2 MB
> - **Total:** ~ 27,574 MB (~26.9 GB)
>
> It is crucial to note that this reported value represents the **theoretical model-intrinsic memory usage**. Real-world deployment incurs additional system overhead (e.g., CUDA context), as highlighted in Fed-pilot [1]. Consequently, the actual memory footprint on edge devices would strictly exceed this value, further **exacerbating** the severity of the memory wall. Given that system overhead is device-dependent, we focuse on reporting the minimum memory required to load the model. We have explicitly clarified these profiling settings in the **caption of Figure 1**.
>
> Furthermore, we fully concur with the reviewer that memory usage scales significantly with increased input token length and batch size. Indeed, we believe this observation **reinforces the motivation** behind our work: since memory consumption already reaches $\sim$26.9 GB under these moderate settings, increasing the batch size or processing longer contexts would render the memory wall an even more prohibitive barrier for edge devices. This makes efficient, memory-saving solutions like ours even more indispensable.
>
>
>
> > **W2 & W3: Memory Usage Formulation & Rationale.**
>
> We are grateful for these constructive feedbacks, as they highlight the central motivation behind our research.
>
> **1.Formal Problem Formulation for Memory Consumption:**
>
> We specifically thank the reviewer for recommending Fed-pilot [1], a reference that offers a critical analysis of memory composition in federated environments. Following Fed-pilot, the total memory consumption $M_{\text{total}}$ can be formulated as:
>
> $$M\_{\text{total}} = \underbrace{M\_{\text{params}} + M\_{\text{optimizer}} + M\_{\text{activations}}}_{\text{Model-Intrinsic Memory}} + \underbrace{M\_{\text{system}}}\_{\text{System Overhead}}$$
>
> Where:
>
> - $M_{\text{params}}$: Memory occupied by model parameters (Static).
> - $M_{\text{optimizer}}$: Memory for optimizer states (e.g., gradients, momentum in Adam).
> - $M_{\text{activations}}$: Memory for intermediate feature maps (Dynamic; depends on batch size/sequence length).
> - $M_{\text{system}}$: Memory for system/CUDA context and workspace (Varies by device/software).
>
> In this paper, we focus exclusively on model-intrinsic memory usage. Since system overhead is device-dependent, variable, and **negligible relative to** the model-intrinsic footprint, we concentrate on optimizing this quantifiable portion of the memory cost. We have incorporated relevant analysis into **Section 1**.
>
> **2.Rationale for Targeting $M_{\text{params}}$:**
>
> We fully agree with the reviewer that total memory comprises multiple components and that the overall memory distribution scales dynamically with batch size and token length. However, our strategic focus on $M_{\text{params}}$ is driven by two critical factors unique to the deployment of LLMs on edge devices:
>
> - **The "Memory Floor" Barrier:** $M_{\text{params}}$ represents the fundamental **"memory floor"**—the static, minimum requirement a device must meet before training can commence. If $M_{\text{params}}$ exceeds a device's physical RAM, optimizing dynamic components (e.g., reducing batch size to minimize $M_{\text{activations}}$) becomes futile, as the model cannot even be loaded into memory.
> - **Dominance of Parameters in LLMs:** Unlike smaller CNNs, for LLMs, parameters constitute the overwhelming majority of the memory footprint. As our profiling in Figure 1(c) demonstrates, under standard fine-tuning settings, $M_{\text{params}}$ accounts for **92.8%** of the model-intrinsic memory. This vastly exceeds the memory consumed by others.
>
> Therefore, FedPruner is strategically designed to tackle this primary bottleneck. By intelligently pruning $M_{\text{params}}$, we effectively lower this "memory floor," enabling the model to fit within the restricted capacity of edge devices. This distinguishes our approach from methods that primarily focus on optimizing $M_{\text{activations}}$ or $M_{\text{optimizer}}$, which remain secondary concerns if the model itself cannot be loaded.

---

> ### Author Response · Authors · 2025-11-21
> **Response to Reviewer Lk6t (2 / 3)**
>
> > **W4: Discussion with Related Works.**
>
> We sincerely thank the reviewer for highlighting these relevant and excellent studies. We apologize for the oversight in our initial submission. However, we wish to clarify the **fundamental distinctions** between these approaches and our work:
>
> 1. **Different Optimization Targets:** Works such as Fed-Pilot [1], Fed-HeLLo [2], FlexLoRA [3], and HETLoRA [4] primarily focus on optimizing the LoRA modules (e.g., rank or layer-wise allocation). As illustrated in Figure 1(c), LoRA modules represent a negligible fraction of the memory footprint (e.g., merely 0.018% for LLaMA2-7B). In contrast, FedPruner targets the base model parameters, effectively addressing the dominant memory bottleneck.
> 2. **The Prerequisite Barrier:** Crucially, these LoRA-centric methods operate under the assumption that the device can load the complete, unmodified base model into memory. As demonstrated in Table 1 and Figure 1(a), this assumption often fails for resource-constrained edge devices, leading to 0% participation for LLaMA2-13B. FedPruner overcomes this fundamental feasibility barrier, enabling participation where baseline methods fail.
>
> **New Experimental Comparison:** To empirically validate the superiority of FedPruner, we conduct additional experiments on LLaMA2-7B. As shown in the table below, FedPruner achieves average performance gains of up to **4.41%** over these baselines. This performance gap stems from the fact that competing approaches cannot effectively leverage valuable data from low-memory devices (due to insufficient memory savings restricting device participation), whereas FedPruner enables broad participation.
>
> **Complementary Nature:** It is important to acknowledge that these excellent works contribute significantly to federated fine-tuning from different perspectives. We view FedPruner as **complementary, rather than competitive**, to these approaches. A logical pipeline could involve using FedPruner to first create a fit-able submodel, and subsequently applying a method like Fed-HeLLo [2] to optimize LoRA allocation within that submodel. We have incorporated relevant discussions and experiments into **Section 5** and **Appendix J.2**.
>
> **Table: Performance comparison with advanced LoRA-level optimization methods on LLaMA2-7B.**
>
> | **Method**    | **TruthfulQA** | **MMLU** | **IFEval** | **BBH** | Average             |
> | ------------- | -------------- | -------- | ---------- | ------- | ------------------- |
> | Fed-pilot[1]  | 44.15%         | 41.83%   | 29.14%     | 38.85%  | **38.49% (-3.50%)** |
> | Fed-HeLLo[2]  | 43.86%         | 41.62%   | 28.71%     | 38.43%  | **38.16% (-3.83%)** |
> | FlexLoRA[3]   | 43.67%         | 41.46%   | 28.17%     | 38.06%  | **37.84% (-4.15%)** |
> | HETLoRA[4]    | 43.42%         | 41.21%   | 27.93%     | 37.76%  | **37.58% (-4.41%)** |
> | **FedPruner** | 49.95%         | 43.64%   | 33.72%     | 40.65%  | __41.99%__          |
>
>
>
> > **W5: Clarification on Metrics and Role of Figure 2.**
>
> We appreciate the reviewer’s constructive comment. We fully concur that perplexity (PPL) alone is insufficient for evaluating performance on complex downstream tasks, and we apologize if the specific objective of Figure 2 was not explicitly stated. We wish to clarify that Figure 2 (and Figure 3) served strictly as a **preliminary, motivational investigation**, rather than a comprehensive performance evaluation. Its sole purpose was to diagnose "model collapse"—demonstrating that simple, heuristic pruning strategies cause a catastrophic spike in PPL, thereby destroying the model's fundamental language modeling capability.
>
> **1. Comprehensive Downstream Evaluation:** Our primary evaluation (Table 1) explicitly avoids PPL and instead employs the comprehensive, task-specific benchmarks and metrics. As detailed in Section 4.1, this includes:
>
> - **Close-ended Benchmarks:** We evaluate across four distinct core capabilities: TruthfulQA (truthfulness), MMLU (knowledge), IFEval (instruction following), and BBH (complex reasoning). For these benchmarks, we use **Accuracy** as the evaluation metric.
>
> - **Open-ended Benchmarks:** We utilize Vicuna-Bench and MT-Bench to assess multi-turn conversational abilities. For these benchmarks, we employ **GPT-4 Scoring** as the evaluation metric.
>
>   The SOTA performance reported in Table 1 validates FedPruner's effectiveness across these complex tasks. We have incorporated descriptions of these metrics into **Section 4.1**.
>
> **2. Alignment with Established Diagnostic Protocols:** Furthermore, our use of PPL is strictly confined to preliminary diagnostic analysis, which aligns with established methodologies in recent literature, such as ShortGPT [5] (ACL 2025). In these works, PPL is widely accepted as a standard proxy for measuring layer importance during structural analysis. Thus, we adopted this metric to ensure methodological consistency with the community's standard practice.

---

> ### Author Response · Authors · 2025-11-21
> **Response to Reviewer Lk6t (3 / 3)**
>
> > __W6: On the Method's Support for Heterogeneity and Its Aggregation Strategy .__
>
> We thank the reviewer for this crucial question.
>
> **1. Support for Homogeneous and Heterogeneous Settings:**
>
> - **Homogeneous Settings:** We explicitly evaluate scenarios where all devices share identical constraints in Section 4.8 (Table 4). Even when all devices are restricted to a strict, uniform memory budget (e.g., utilizing only **25%** of the full model size), FedPruner outperforms the theoretical LoRA baseline by 1.98% on LLaMA2-13B. This validates its effectiveness in uniform, constrained environments.
> - **Heterogeneous Settings:** Our primary evaluation (Table 1) was conducted under significant resource heterogeneity. As detailed in Appendix E.2 (Line 965), we simulated a realistic environment where device memory budgets were randomly allocated between 3GB and 9GB. FedPruner naturally adapts to this diversity because each client independently and dynamically constructs its own submodel based on its local memory budget and data distribution. This per-client, data-driven adaptation is precisely what enables our method to handle extreme heterogeneity.
>
> **2. Heterogeneous Aggregation Strategy:** We design a robust, layer-wise aggregation mechanism (Appendix B). The process operates as follows:
>
> - The server maintains the full set of global LoRA parameters.
>
> - Participating devices only train and upload parameters for their specific submodels.
>
> - The server performs aggregation on a per-layer basis.
>
> - Here is a concrete example to illustrate the aggregation process. Consider a global model with six layers, $\Theta$ = {$\theta_1$, ..., $\theta_6$}. In round $r$, two devices are selected: Device 1 updates the layers {$\theta_1$, $\theta_3$, $\theta_5$}; Device 2 updates the layers {$\theta_2$, $\theta_3$, $\theta_5$}. The overall aggregation process at the server is performed as follows:
>
>   - **Overlapping Layers ($\theta_3, \theta_5$):** For layers updated by both devices, the server computes a weighted aggregation of the updates from both devices.
>
>   - **Exclusive Layers ($\theta_1, \theta_2$):** For layers updated by a single device, the update is applied directly.
>
>   - **Unselected Layers ($\theta_4, \theta_6$):** Layers not selected by any device remain unchanged.
>
> This mechanism effectively accommodates heterogeneous updates. Furthermore, we provide a rigorous convergence analysis in Appendix D to validate the theoretical soundness of this partial update scheme. We have incorporated this concrete example into **Appendix B**.
>
>
>
> ---
>
>
>
> Again, we thank the reviewer for the valuable feedback. Please let us know if there are any other questions or suggestions.
>
>
>
> **References**:
>
> [1] Fed-pilot: Optimizing LoRA Allocation for Efficient Federated Fine-Tuning with Heterogeneous Clients. ArXiv 2024.
>
> [2] Fed-HeLLo: Efficient Federated Foundation Model Fine-Tuning with Heterogeneous LoRA Allocation. IEEE TNNLS 2025.
>
> [3] Federated fine-tuning of large language models under heterogeneous tasks and client resources. NeurIPS 2024.
>
> [4] Heterogeneous LoRA for Federated Fine-tuning of On-Device Foundation Models. EMNLP 2024.
>
> [5] ShortGPT: Layers in Large Language Models are More Redundant Than You Expect. ACL 2025.

---

> ### Author Response · Authors · 2025-11-27
>
> Dear Reviewer Lk6t,
>
> As the discussion phase is entering its final stage, we want to kindly follow up to ensure that our response has adequately addressed your concerns.
>
> We truly value the time you have dedicated to reviewing our work. If there are any remaining questions or if further clarification is needed, please let us know—we are eager to engage in further discussion to improve the paper.
>
> Thank you again for your constructive feedback.
>
> Best regards,
>
> The Authors

---

### Official Review · Reviewer_qN12 · 2025-10-31

**Soundness:** 1
**Presentation:** 3
**Contribution:** 3
**Rating:** 4
**Confidence:** 4

**Summary:**

The paper addresses the challenge of performing federated fine-tuning under memory constraints on resource-limited devices. The authors propose a pruning-based method that strategically removes redundant layers to reduce model parameters and enable large model fine-tuning across heterogeneous clients.

Overall, I appreciate the problem this paper aims to solve, and the proposed method appears to be novel. However, I still have concerns about the soundness of the method design and the adequacy of the experimental evaluation. I hope the author can solve all my confusion in the rebuttal.

**Strengths:**

1. The motivation of the paper is reasonable, and Figure 1 presents it effectively. The excessive memory consumption of model parameters is indeed a key challenge in federated fine-tuning, and the authors have clearly illustrated this issue.

2. This paper is well-written and easy to follow. The overall logic and structure are coherent, and the presentation is clear.

3. The experimental results clearly demonstrate the effectiveness of the proposed method compared with the baselines.

**Weaknesses:**

1. The authors assume that resource-constrained devices cannot load the full model parameters. However, Step 1 states that each device performs inference to calculate inter-layer similarity, which inherently requires loading the entire model. This design contradicts the initial assumption about limited device capacity. The authors should clarify how Step 1 can be executed on devices that are unable to host the full model.

2. Steps 1-3 appear to introduce considerable computational overhead, which may be unacceptable for resource-constrained devices. The authors should provide a detailed analysis or justification regarding this issue.

3. The paper lacks discussion and comparison with traditional heterogeneous federated learning methods such as HeteroFL [R1], DepthFL [R2], and AdaptiveFL [R3]. It would be important to clarify whether these conventional approaches can be applied to federated fine-tuning. If they can, the authors should include corresponding baselines for comparison; if not, they should provide explanations and discussions on why these methods are not applicable.

4. The paper lacks experiments evaluating the adaptability of the proposed method to different types of models. It would be valuable to assess how the method performs across various model architectures to better demonstrate its generalization capability.

5. The paper lacks evaluation on the basic settings of federated learning, such as varying the number of clients, the participation ratio of clients in each round, and the distribution of client capabilities. These analyses are important to demonstrate the robustness and scalability of the proposed method.


[R1] Heterofl: Computation and communication efficient federated learning for heterogeneous clients

[R2] Depthfl: Depthwise federated learning for heterogeneous clients

[R3] AdaptiveFL: Adaptive heterogeneous federated learning for resource-constrained AIoT systems

**Questions:**

Please see weaknesses.

---

> ### Author Response · Authors · 2025-11-21
> **Response to Reviewer qN12 (1 / 3)**
>
> We sincerely thank you for your insightful and constructive review. We have carefully considered each point you raised and have incorporated corresponding revisions and clarifications into the uploaded revised manuscript. Below are our point-by-point responses to your concerns.
>
>
>
> > **W1: Clarification on the Feasibility of Step 1 Given Memory Constraints.**
>
> We sincerely apologize for not making this process clearer in the original submission. You are entirely correct: a resource-constrained device cannot load the full model simultaneously. We wish to clarify that Step 1 (Layer Similarity Computation) does not require loading the entire model. Instead, we employ an efficient **pipelined, layer-by-layer inference workflow**, ensuring the process always operates within the device’s memory budget. This process works as follows:
>
> 1. **Adaptive Chunk Loading:** The device loads only the first chunk of layers that fits within its memory budget (e.g., layers 1-3, if its memory can only afford three layers at a time).
> 2. **Compute, Cache, and Release:** A forward pass is performed on this chunk using a single data batch. Once the necessary output activations are cached and stored, the device immediately offloads the layer parameters to free memory.
> 3. **Sequential Iteration:** The device then loads the subsequent chunk (e.g., layers 4-6), utilizing the cached activations from the previous block as input. This cycle repeats until the forward pass is complete for all layers.
>
> This pipelined workflow strictly bounds the memory footprint of the CKA calculation to the device's available budget. By adaptively scaling to operate within specific hardware constraints, our method **eliminates the prerequisite** of loading the full model, thereby resolving any perceived contradiction in our design. We have included a detailed description of this workflow in **Appendix L**.
>
>
>
> > **W2: Computational Overhead and Feasibility of Steps 1-3 on Resource-Constrained Devices.**
>
> We understand the reviewer's concern regarding the computational overhead associated with the initial setup steps. We wish to respectfully clarify that the cost introduced by Steps 1-3 is **minimal and negligible**. Here is a detailed breakdown of the costs:
>
> - **Step 1 (Layer Similarity Computation):** This step's overhead is dominated by a single forward pass using only one batch of data. This one-time cost is computationally insignificant relative to the local fine-tuning process. For instance, for LLaMA2-7B, with a batch size of 16, this single forward pass incurs an overhead of approximately ~54.3 TFLOPs. A single local fine-tuning round requires approximately ~2232.0 TFLOPs. Therefore, the setup cost for Step 1 represents only a **minor fraction of local computation (2.4%)**.
> - **Steps 2-3 (Graph Construction & Partitioning):** These steps operate on a small $N \times N$ matrix (where $N$ is the number of layers), not the high-dimensional parameter space. For a model like LLaMA2-7B ($N=32$), this requires partitioning a small $32 \times 32$ matrix. The computational cost of this operation is measured in milliseconds, making it computationally insignificant for edge devices.
>
> Moreover, the minimal, one-time computational cost introduced by Steps 1-3 is vastly outweighed by the substantial **efficiency gains** achieved by fine-tuning a smaller, pruned submodel. Our results in Table 3 underscore FedPruner's high efficiency, demonstrating up to a 2.33x convergence speedup. A detailed discussion has been added to **Appendix K.**

---

> ### Author Response · Authors · 2025-11-21
> **Response to Reviewer qN12 (2 / 3)**
>
> > **W3: Applicability and Comparison of Traditional Heterogeneous FL Methods.**
>
> We thank the reviewer for highlighting these relevant references. We wish to clarify that these traditional methods are generally unsuitable for federated fine-tuning of LLMs for two primary reasons.
>
> 1. **Different Problem Paradigms:** These methods are designed for training smaller models (e.g., CNNs) from scratch. This objective is distinct from our focus on the federated fine-tuning of LLMs, which presents unique challenges related to model scale and architecture.
> 2. **Methodological Inapplicability:** Many of these methods are not directly compatible with the Transformer architecture. For instance, techniques like HeteroFL [1] and AdaptiveFL [3] rely on mechanisms such as channel pruning, which is specific to CNNs and fundamentally incompatible with LLMs.
>
> DepthFL [2] is the most relevant, as it allocates submodels of varying depths to address resource constraints. However, its heuristic approach is ill-suited for the LLM fine-tuning scenario. Its strategy can lead to severe training imbalance, as critical deep layers may be trained only by a few high-capacity devices, **or even none at all**. We conduct new experiments to empirically validate this. As shown in the table below, on LLaMA2-7B, DepthFL incurs a significant 5.21% average performance loss. This is because only a minority of devices can train the deep layers, leading to insufficient training, a problem directly solved by FedPruner's systematic, importance-aware layer selection. A detailed discussion has been included in **Appendix K**.
>
> Moreover, we would also like to respectfully note that our paper already includes comparisons against the direct SOTA in memory-heterogeneous federated fine-tuning, namely FLoRA[4] (NeurIPS 2024) and FedRA[5] (ECCV 2024).
>
> **Table: Performance comparison with DepthFL on TinyLLaMA and LLaMA2-7B.**
>
> | Model     | **Method**    | **TruthfulQA** | **MMLU** | **IFEval** | **BBH** | Average             |
> | --------- | ------------- | -------------- | -------- | ---------- | ------- | ------------------- |
> | TinyLLaMA | DepthFL       | 37.72%         | 24.85%   | 16.14%     | 26.00%  | __26.18% (-2.15%)__ |
> |           | **FedPruner** | 39.16%         | 26.18%   | 19.87%     | 28.09%  | __28.33%__          |
> | LLaMA2-7B | DepthFL       | 42.31%         | 40.96%   | 26.52%     | 37.34%  | __36.78% (-5.21%)__ |
> |           | **FedPruner** | 49.95%         | 43.64%   | 33.72%     | 40.65%  | __41.99%__          |
>
>
>
> > **W4: Evaluating Generalizability Across Different Model Architectures.**
>
> We appreciate the reviewer’s insightful suggestion regarding the need to demonstrate adaptability across diverse model architectures. To address this comprehensively, we extend our evaluation to include widely-used SOTA LLMs spanning different families and parameter scales: LLaMA3.2-3B, LLaMA3.1-8B, and Qwen2.5-7B.
>
> The results presented in the table below confirm that FedPruner significantly and consistently outperforms the theoretical (LoRA) baseline across all models. It achieves average performance gains of **+8.62% on LLaMA3.2-3B**, **+8.11% on LLaMA3.1-8B**, and **+5.15% on Qwen2.5-7B**. These findings substantiate that FedPruner is not confined to specific LLaMA variants but offers a scalable solution across diverse Transformer-based models, thereby validating its strong generalization capability. We have incorporated these new results into **Section 4.9**.
>
> **Table: Performance evaluation on different model architectures**.
>
> | Model           | Method             | TruthfulQA | MMLU   | IFEval | BBH    | Average             |
> | --------------- | ------------------ | ---------- | ------ | ------ | ------ | ------------------- |
> | **LLaMA3.2-3B** | LoRA (Theoretical) | 44.25%     | 54.19% | 53.60% | 46.84% | __49.72%__          |
> |                 | **FedPruner**      | 49.62%     | 60.43% | 66.58% | 56.71% | __58.34%(+8.62%)__  |
> | **LLaMA3.1-8B** | LoRA (Theoretical) | 48.07%     | 63.31% | 47.32% | 62.69% | __55.35%__          |
> |                 | **FedPruner**      | 53.98%     | 67.95% | 61.40% | 70.51% | __63.46% (+8.11%)__ |
> | __Qwen2.5-7B__  | LoRA (Theoretical) | 48.02%     | 41.89% | 33.68% | 42.63% | **41.56%**         |
> |                 | **FedPruner**      | 54.65%     | 46.70% | 39.51% | 45.96% | __46.71% (+5.15%)__ |

---

> ### Author Response · Authors · 2025-11-21
> **Response to Reviewer qN12 (3 / 3)**
>
> > **W5: Evaluating Robustness and Scalability under Varying FL Settings.**
>
> We appreciate the reviewer’s constructive suggestion regarding the evaluation of fundamental FL settings. In response, we offer the following clarifications and present new comprehensive sensitivity analyses to address these concerns.
>
> **1. Robustness to Client Capability Distribution:** We respectfully clarify that this aspect is already addressed in our main paper. Section 4.8 (Table 4), titled "Performance under different memory constraints," was specifically designed to evaluate adaptability to heterogeneous client capabilities. By varying the available memory budget from 25% (extreme constraint) to 75% of the global model size, this experiment directly demonstrates FedPruner's superior adaptability. The results show a significant improvement, such as +1.98% gain over the theoretical baseline at the strict 25% budget on LLaMA2-13B, validating our method's robustness against severe memory heterogeneity.
>
> **2.Scalability with Varying Number of Clients:** To further rigorously assess scalability, we conduct new experiments on TinyLLaMA by varying the total number of clients across {10, 20, 30, 50}. As shown in the table below, FedPruner remains robust across different population sizes, consistently maintaining a performance advantage over FedIT. Notably, with a larger client population of 50, FedPruner achieves an average accuracy improvement of **2.06%**.
>
> **Table: Performance evaluation with varying numbers of Clients.**
>
> | Number of Clients | Method        | TruthfulQA | MMLU   | IFEval | BBH    | Average             |
> | ----------------- | ------------- | ---------- | ------ | ------ | ------ | ------------------- |
> | 10                | FedIT         | 38.42%     | 25.38% | 17.65% | 26.59% | **27.01%**          |
> |                   | __FedPruner__ | 39.71%     | 26.53% | 21.44% | 28.75% | **29.11% (+2.10%)** |
> | 20                | FedIT         | 37.87%     | 24.94% | 16.19% | 26.03% | __26.26%__          |
> |                   | __FedPruner__ | 39.16%     | 26.18% | 19.87% | 28.09% | __28.33% (+2.07%)__ |
> | 30                | FedIT         | 37.78%     | 24.85% | 16.15% | 26.00% | **26.20%**          |
> |                   | __FedPruner__ | 39.05%     | 26.11% | 19.76% | 28.02% | **28.24% (+2.04%)** |
> | 50                | FedIT         | 37.74%     | 24.76% | 16.08% | 25.94% | **26.13%**          |
> |                   | __FedPruner__ | 39.10%     | 26.09% | 19.61% | 27.95% | **28.19% (+2.06%)** |
>
> **3.Robustness to Client Participation Ratios:** We further investigate the impact of client participation rates by testing ratios of {10\%, 20\%, 50\%} on TinyLLaMA. FedPruner demonstrates remarkable resilience, consistently maintaining high accuracy and outperforming the baseline across all settings. Notably, even under a sparse participation rate of 10%, FedPruner achieves an average performance improvement of 2.07%.
>
> These comprehensive analyses substantiate the robustness and scalability of FedPruner under diverse FL conditions. We have incorporated these results into **Appendix J.4** and **Appendix J.5**.
>
> **Table: Performance evaluation with varying participation ratios**.
>
> | Participation Ratio | Method        | TruthfulQA | MMLU   | IFEval | BBH    | Average              |
> | ------------------- | ------------- | ---------- | ------ | ------ | ------ | -------------------- |
> | 10%                 | FedIT         | 37.87%     | 24.94% | 16.19% | 26.03% | __26.26%__           |
> |                     | __FedPruner__ | 39.16%     | 26.18% | 19.87% | 28.09% | __28.33% (+2.07%)__  |
> | 20%                 | FedIT         | 38.45%     | 25.50% | 17.72% | 26.60% | **27.07%**           |
> |                     | __FedPruner__ | 39.72%     | 26.71% | 21.58% | 28.79% | **29.20% (+2.13%)** |
> | 50%                 | FedIT         | 38.63%     | 25.54% | 18.13% | 26.88% | **27.30%**           |
> |                     | __FedPruner__ | 39.90%     | 27.04% | 22.06% | 28.96% | **29.49% (+2.19%)**  |
>
>
>
> ---
>
> Again, we thank the reviewer for the valuable feedback. Please let us know if there are any other questions or suggestions.
>
>
>
> **References**:
>
> [1] Heterofl: Computation and communication efficient federated learning for heterogeneous clients. 2021, ICLR.
>
> [2] Depthfl: Depthwise federated learning for heterogeneous clients. 2024, ICLR.
>
> [3] AdaptiveFL: Adaptive heterogeneous federated learning for resource-constrained AIoT systems. 2024, DAC.
>
> [4] Flora: Federated fine-tuning large language models with heterogeneous low-rank adaptations. 2024, NeurIPS.
>
> [5] Fedra: A random allocation strategy for federated tuning to unleash the power of heterogeneous clients. 2024, ECCV.

---

> > ### Comment · Reviewer_qN12 · 2025-11-25
> >
> > Thank you for your response and the clarifications provided. However, I still have several concerns that I hope the authors can address:
> >
> > C1 (Regarding W1): While I understand the proposed pipeline process, it appears that the standard model training procedure could also adopt a similar pipeline-with-caching strategy to overcome memory constraints. The authors need to further clarify whether their specific pipeline approach introduces any additional overhead or costs compared to such a potential baseline.
> >
> > C2 (Regarding W3): I maintain my disagreement regarding the characterization of fine-tuning versus training from scratch as a fundamental paradigm difference. The authors should provide a clearer justification for this distinction and elaborate on the unique challenges that arise specifically in their setting. Furthermore, while I acknowledge the architectural differences between Transformers and CNNs, it would be valuable to know if techniques such as pruning FNN channels or attention heads, similar to methods used in HeteroFL and AdaptiveFL, could be implemented within the proposed framework. A more detailed analysis on this point is needed.
> >
> > C3 (Regarding W5):
> >
> > * Different Client Capability Distribution: My original point was to consider a scenario with, for instance, 100 devices, each assigned a different memory budget. A concrete example would be: 50 clients with only 25% of the memory budget, 25 clients with 50%, and the remaining 25 clients with 75%, rather than all devices sharing the same memory budget.
> >
> > * Different Client Numbers: My concern pertains to cross-device scenarios with varying scales of participation, such as comparing situations with 100 clients or 200 clients.
> >
> > In addition, I strongly recommend that the authors supplement their scalability experiments with comparative results against recent and relevant baseline methods.

---

> > > ### Author Response · Authors · 2025-11-27
> > > **Response to Follow-up Comments from Reviewer qN12 (1 / 2)**
> > >
> > > We sincerely thank the reviewer for the prompt and engaging follow-up.
> > >
> > >
> > >
> > > > **C1: Exploring Caching Methods for Standard Training.**
> > >
> > > We agree that a "pipeline-with-caching" strategy effectively reduces memory usage in standard training. However, applying such strategies to federated fine-tuning encounters two fundamental limitations: ineffectiveness and prohibitive I/O overhead.
> > >
> > > **1. Limitations of Existing Caching Techniques.** Representative approaches include Gradient Checkpointing [1], which selectively caches activations and recomputes the rest during backward passes, and Gradient Cache [2], which decomposes batches into micro-batches to simulate large-batch training. However, these methods primarily optimize intermediate activations, ignoring the dominant memory bottleneck in LLMs: model parameters (e.g., 92.8% for LLaMA2-7B). Consequently, such activation-centric techniques are insufficient to resolve the memory constraints.
> > >
> > > **2. Infeasibility of Pipeline-Based Training.** Directly applying our pipeline strategy to standard training is practically infeasible. This incurs prohibitive I/O overhead, as it necessitates swapping parameters in and out for every step.
> > >
> > > - **Bi-directional I/O Cost:** Unlike inference, training necessitates redundant parameter swapping: chunks must be loaded/offloaded for the forward pass and re-loaded (in reverse) for backpropagation. This repetitive read-write cycle saturates the system's I/O bandwidth.
> > > - **Cumulative Latency in FL:** This bottleneck is exacerbated in FL. Given the hundreds of rounds and local iterations, the cumulative latency from frequent storage-GPU transfers causes total training time to scale exponentially.
> > >
> > > **3. Efficiency of FedPruner.** FedPruner strategically restricts the pipeline mechanism to the initialization phase to circumvent I/O bottlenecks. The pipeline is utilized exclusively for a single forward pass, eliminating the need for backward passes or repetitive reloading. As demonstrated, this initialization incurs a negligible cost (merely 2.4% of total computation).
> > >
> > >
> > >
> > > > **C2: Paradigm Distinction & Width Pruning.**
> > >
> > > We appreciate the opportunity to deepen this discussion.
> > >
> > > **1. Paradigm Distinction.** We apologize for the confusion caused by our previous terminology. We wish to clarify that the distinction between "training from scratch" and "fine-tuning" is fundamental because it dictates the model's adaptability:
> > >
> > > - **Training from Scratch (High Plasticity):** Methods like HeteroFL operate in a high-plasticity regime. With randomly initialized weights, the model actively learns to fit the constrained architecture. For example, if a client is restricted to a subset of channels, the optimization process naturally drives the model to consolidate essential features into those specific indices. This constitutes an active process of knowledge construction.
> > > - **LLM Fine-Tuning (Low Plasticity):** This involves adapting a pre-trained model where knowledge is entangled across the full width of embedding dimensions and attention heads. In this low-plasticity regime, the model lacks sufficient data to "re-learn" or "re-map" global knowledge into a subset of channels. This fundamentally represents a process of knowledge adaptation.
> > > - **Unique Challenge:** Therefore, applying "width slicing" to a pre-trained LLM ruptures the semantic integrity of feature vectors, leading to catastrophic forgetting and representational collapse.
> > >
> > > **2. Applicability of Width Pruning.** We agree that pruning specific Attention Heads or FNN channels is technically feasible. However, implementing this introduces severe bottlenecks:
> > >
> > > - **Information Bottleneck:** Width pruning results in a "Narrow and Deep" topology, which reduces the dimensionality of the semantic space by forcing high-dimensional representations into lower-dimensional projections. This causes irreversible information loss at every layer and compromises the model's structural integrity.
> > > - **Matrix Mismatch & Aggregation:** Pruning FFN channels or Heads creates irregular, dimensionally mismatched matrices across clients. This heterogeneity renders standard aggregation mathematically intractable, requiring complex mechanisms to align disparate sub-structures.
> > > - **Superiority of Layer Pruning:** FedPruner maintains a "Wide and Shallow" topology. Leveraging the residual nature of Transformers ($x_{l+1} = x_l + F(x_l)$), removing a layer reduces to an identity mapping. This preserves the high-dimensional "width" of the information highway, allowing pre-trained features to propagate intact.
> > >
> > > **3. FedPruner+: Granularity without Destruction.** To address the need for granularity, FedPruner+ (Section 3.3) treats MHA and FFN as independent prunable units, achieving the flexibility of HeteroFL-style methods without breaking matrix integrity. This enables the discovery of efficient, heterogeneous architectures (Figure 7) while preserving the pre-trained structural priors.

---

> > > ### Author Response · Authors · 2025-11-27
> > > **Response to Follow-up Comments from Reviewer qN12 (2 / 2)**
> > >
> > > > **C3: Robustness & Scalability.**
> > >
> > > We thank the reviewer for specifying the desired experimental settings. In response, we have conducted extensive experiments to rigorously demonstrate the robustness and scalability of FedPruner.
> > >
> > > **1. Robustness to Different Capability Distribution.** We simulate a realistic heterogeneous environment following the suggested setting: 50 (25% Memory), 25 (50% Memory), and 25 (75% Memory) clients. FedPruner significantly outperforms all baselines.
> > >
> > > - **Memory-Unaware Methods (FedIT, DoFIT, FedSA-LoRA)**: These methods fail completely. Since no device can hold the full model (0% participation), the model degenerates to its Zero-Shot performance.
> > > - **Insufficiently Memory-Aware Methods (FLoRA, FwdLLM)**: Although FLoRA and FwdLLM optimize LoRA ranks and activations, these components represent a minor fraction of the total memory (~0.012% and ~4.8%). Consequently, they also fail to address the memory constraints.
> > > - **Heuristic Memory-Aware Methods (FedRA)**: While FedRA enables participation via random pruning, this stochastic approach degrades the model's representation capability, resulting in a 3.93% average performance gap. In contrast, FedPruner intelligently constructs tailored submodels for each client, effectively orchestrating training across heterogeneous devices.
> > >
> > > **Table: Performance comparison with mixed client capability on LLaMA2-13B (INT4).**
> > >
> > > | **Method**                 | **TruthfulQA** | **MMLU** | **IFEval** | **BBH** | Average         |
> > > | -------------------------- | -------------- | -------- | ---------- | ------- | --------------- |
> > > | FedIT & DoFIT & FedSA-LoRA | 42.83%         | 49.65%   | 30.35%     | 41.03%  | 40.97% (-7.45%) |
> > > | FLoRA & FwdLLM             | 42.83%         | 49.65%   | 30.35%     | 41.03%  | 40.97% (-7.45%) |
> > > | FedRA                      | 47.86%         | 52.11%   | 35.44%     | 42.56%  | 44.49% (-3.93%) |
> > > | **FedPruner**              | 52.36%         | 54.87%   | 41.94%     | 44.52%  | **48.42%**      |
> > >
> > > **2.Scalability with Varying Client Numbers.** To address concerns regarding scalability against recent baselines, we perform two sets of expanded experiments.
> > >
> > > - **Comparative Evaluation (50 Clients).** We extend our previous scalability analysis to include a full suite of SOTA baselines. As detailed below, FedPruner consistently outperforms all competing methods, improving average accuracy by up to 2.06%.
> > >
> > > **Table: Performance evaluation with 50 Clients on TinyLLaMA.**
> > >
> > > | Method        | TruthfulQA | MMLU   | IFEval | BBH    | Average         |
> > > | ------------- | ---------- | ------ | ------ | ------ | --------------- |
> > > | FedIT         | 37.74%     | 24.76% | 16.08% | 25.94% | 26.13% (-2.06%) |
> > > | DoFIT         | 38.66%     | 25.48% | 18.60% | 27.31% | 27.51% (-0.68%) |
> > > | FedSA-LoRA    | 38.47%     | 25.41% | 18.89% | 27.39% | 27.54% (-0.65%) |
> > > | FLoRA         | 38.34%     | 25.17% | 17.46% | 27.12% | 27.02% (-1.17%) |
> > > | FwdLLM        | 38.38%     | 25.33% | 18.51% | 27.20% | 27.36% (-0.83%) |
> > > | FedRA         | 38.56%     | 25.34% | 19.10% | 27.41% | 27.60% (-0.59%) |
> > > | __FedPruner__ | 39.10%     | 26.09% | 19.61% | 27.95% | **28.19%**      |
> > >
> > > - **Massive-Scale Evaluation.** To further validate the scalability beyond the 100-client setting discussed above, we significantly expand the client population. Given the data volume constraints of Alpaca-GPT4, we experiment on text classification, spanning 100 (20NEWS), 1,000 (AGNEWS, YELP-P), and 10,000 (YAHOO) clients under heterogeneous memory constraints. FedPruner consistently achieves superior performance with average gains of up to 11.56%. These results strongly substantiate FedPruner's scalability across massive client populations and robustness to diverse hardware limits.
> > >
> > > **Table: Performance comparison on text classification tasks with DistilBERT.**
> > >
> > > | Method        | YELP-P | AGNEWS | YAHOO  | 20NEWS | Average          |
> > > | ------------- | ------ | ------ | ------ | ------ | ---------------- |
> > > | FedIT         | 77.48% | 82.26% | 62.87% | 67.35% | 72.49% (-9.55%)  |
> > > | DoFIT         | 77.73% | 83.34% | 63.51% | 68.64% | 73.31% (-8.73%)  |
> > > | FedSA-LoRA    | 77.64% | 83.19% | 63.35% | 68.26% | 73.11% (-8.93%)  |
> > > | FLoRA         | 77.52% | 82.93% | 63.12% | 67.91% | 72.87% (-9.17%)  |
> > > | FwdLLM        | 77.61% | 83.03% | 61.93% | 59.34% | 70.48% (-11.56%) |
> > > | FedRA         | 77.91% | 84.25% | 65.03% | 72.71% | 74.98% (-7.06%)  |
> > > | __FedPruner__ | 85.36% | 91.23% | 71.34% | 80.23% | **82.04%**       |
> > >
> > >
> > >
> > > We once again thank you for your time and constructive comments. We hope that these clarifications and additional experiments can effectively address your concerns.
> > >
> > >
> > >
> > > ---
> > >
> > >
> > >
> > > **References:**
> > >
> > > [1] Melon: Breaking the memory wall for resourceefficient on-device machine learning. 2022 MobiSys.
> > >
> > > [2] Scaling deep contrastive learning batch size under memory limited setup. 2021 Arxiv.

---

### Official Review · Reviewer_Cret · 2025-10-31

**Soundness:** 2
**Presentation:** 3
**Contribution:** 2
**Rating:** 4
**Confidence:** 4

**Summary:**

This paper proposes FEDPRUNER, a framework designed to address the memory limitations that prevent resource-constrained devices from participating in federated fine-tuning of large language models. The authors develop a macro–micro synergistic pruning framework. Experiments on NLP tasks using LLaMA-based models show that the proposed pruning strategy effectively reduces memory consumption while maintaining model performance.

**Strengths:**

1. This paper addresses a highly relevant and timely problem. The memory bottleneck could prevent resource-constrained devices from participating in federated fine-tuning of LLMs.

2. The proposed macro–micro synergistic pruning framework introduces a novel perspective for coordinating layer selection at multiple granularity levels.

3. The experimental evaluation includes comprehensive baselines covering both memory-unaware and memory-aware methods, and the results appear promising and consistent across benchmarks.

**Weaknesses:**

1. This paper’s motivation lacks clear logical grounding due to missing details and several reasoning gaps.
a. The discussion of the LLaMA-related OOM issue (lines 37–39, Fig. 1(a)) omits essential information such as data precision or quantization settings, which critically affect actual memory usage.
b. The performance discrepancy between theoretical and empirical results in Fig. 1(b) when using TinyLLaMA is not analyzed or explained.
c. Section 2.3 suffers from unclear reasoning and logical jumps. For instance, why must the “memory wall” problem (lines 120–122) necessarily be addressed through pruning? Are there alternative solutions? Moreover, the rationale for multi-layer pruning and the specific choice of pruning ten layers (lines 136–141) is not justified. The paper does not clearly explain why the heuristic baselines underperform or how the proposed method conceptually builds upon and extends these heuristics.

2. The proposed method appears to be a general pruning strategy, and its specific connection to the federated learning setting is not clearly articulated. Moreover, if each client applies FEDPRUNER with different pruning configurations, would the aggregation process become highly complex? The paper would benefit from a deeper analysis of how the proposed framework handles such structural heterogeneity and its implications for global model aggregation and convergence.

3. The experimental comparison suffers from missing and potentially unfair baselines. No other pruning-based methods are included in baselines, making it difficult to assess the effectiveness of FEDPRUNER’s pruning strategy relative to existing state-of-the-art pruning approaches for federated LLMs. Moreover, all compared baselines perform full-model fine-tuning, whereas FEDPRUNER fine-tunes pruned sub-models with inherently lower training time, memory usage, and communication costs, which compromises the fairness of the comparison.

4. The core experiments focus solely on the LLaMA family and the NLP task domain. Given the rapid evolution of open LLMs, the use of LLaMA 2-7B appears somewhat outdated. It is recommended to include evaluations on more recent models, such as LLaMA 3, Gemma 3, or Qwen 2.5, to enhance the relevance and robustness of the findings. Moreover, the applicability of the proposed approach to other generative or multimodal tasks remains to be validated.

5. The related work section overlooks several key studies that are closely related to FEDPRUNER. For instance, FlexLoRA [1] addresses heterogeneous rank adaptation in federated fine-tuning, LEGENDS [2] focuses on managing memory and computational heterogeneity during real-world deployment, and dynamic parameter pruning methods such as FedMef [3] explore similar resource-efficient objectives. A deeper discussion of how FEDPRUNER differs from or builds upon these works would help better position the contribution.

[1] Bai et al. Federated Fine-tuning of Large Language Models under Heterogeneous Tasks and Client Resources. NuerIPS’2024.
[2] Liu et al. Adaptive Parameter-Efficient Federated Fine-Tuning on Heterogeneous Devices. IEEE TMC’2025.
[3] Huang et al. FedMef: Towards Memory-efficient Federated Dynamic Pruning. CVPR’2024.

**Questions:**

See weak points.

---

> ### Author Response · Authors · 2025-11-21
> **Response to Reviewer Cret (1 / 3)**
>
> We sincerely thank you for your insightful and constructive review. We have carefully considered each point you raised and have incorporated corresponding revisions and clarifications into the uploaded revised manuscript. Below are our point-by-point responses to your concerns.
>
>
>
> > __W1: Clarifications on Logical Grounding and Experimental Details.__
>
> - **(a) Experimental Details in Fig. 1(a):** We apologize for any ambiguity. We have updated the caption of Fig.1(a) to explicitly state that the memory analysis uses unquantized model parameters (FP16/FP32), with a batch size of 16 and a maximum sequence length of 512.
> - **(b) TinyLLaMA Performance Discrepancy in Fig. 1(b):** Sorry for the confusion. This discrepancy is not an anomaly but central evidence of the memory wall. The gap between "Practical" and "Theoretical" stems from hardware limitations.
>   - **Memory Constraints:** Edge devices possess heterogeneous memory, ranging from 3GB to 9GB (Lines 965-971). Fig. 1(a) shows that TinyLLaMA requires 5.4GB, exceeding the budget of low-end devices (e.g., 3GB).
>   - **Participation Failure:** Consequently, these memory-constrained devices are precluded from participating, resulting in a participation rate of only 60% (Tab. 1).
>   - **Performance Drop**: This inability to leverage data from all clients causes the observed performance drop. We have clarified this causal chain in **Section 2.2**.
>
> - **(c) Rationale for Pruning & "10-Layer" Experiment:**
>
>   - **Why Pruning?** Our adoption of layer pruning is empirically driven by the bottleneck analysis in Fig. 1(c), which reveals that model parameters dominate memory consumption (92.8% for LLaMA2-7B), far outweighing activations (7.2%) and LoRA (0.018%). Unlike methods targeting these minor components (e.g., FwdLLM[1], FLoRA[2]), layer pruning directly resolves the root memory constraint. Moreover, it offers superior computational efficiency compared to knowledge distillation and orthogonal to quantization (Line 969).
>
>   - **Why Prune 10 Layers?** This is a motivational stress test, not a hyperparameter choice. Its purpose is to demonstrate that at the aggressive compression rates required by real devices (e.g., fitting 13B on a 3GB device), existing heuristics fail catastrophically (evidenced by the high PPL in Fig. 3).
>
>   - **How FedPruner Builds on This:** FedPruner addresses the lack of principled functional preservation in prior works by replacing static heuristics with a robust, data-driven framework. By integrating FDLO (Sec. 3.1) and IALS (Sec. 3.2), we systematically identify critical layers to construct optimal, device-specific submodels. This design is substantiated by ablation study (Tab. 2), where removing them results in a 4.30% degradation on LLaMA2-13B.
>
> > **W2: Connection to FL and Handling of Structural Heterogeneity.**
>
> We appreciate the reviewer’s insightful question.
>
> - **FL-Specific Design vs. General Pruning:** FedPruner fundamentally diverges from general pruning [3] [4] (which yields a single, static model for all uses) by explicitly addressing FL heterogeneity:
>
>   - **Resource Heterogeneity:** Unlike fixed-size pruning, FedPruner enables client-specific construction, where the submodel structure (e.g., the number of layers) is strictly dictated by client memory budgets (Lines 196-200).
>   - **Data Heterogeneity:** The pruning is data-driven. Clients utilize their local data to compute inter-layer similarities (Lines 238-240), which guide the layer orchestration process. This ensures the submodel is optimized for the client's data distribution.
>
>   This per-client, data-driven, and memory-aware adaptation constitutes the core of FedPruner. We provide a detailed discussion in **Appendix K**.
>
> - **Aggregation of Heterogeneous Structures:** We design a streamlined, per-layer aggregation mechanism to handle heterogeneous updates (Appendix B). The process is as follows:
>
>   - The server maintains the full set of LoRA parameters.
>
>   - Devices only train and upload parameters for their specific submodels.
>
>   - The server performs aggregation on a per-layer basis.
>
>   - **Illustrative Example:** Consider a global model with six layers, $\Theta$ = {$\theta_1$, ..., $\theta_6$}. In round $r$, two devices are selected: Device 1 updates the layers {$\theta_1$, $\theta_3$, $\theta_5$}; Device 2 updates the layers {$\theta_2$, $\theta_3$, $\theta_5$}. The aggregation is performed as:
>
>     - **Overlapping Layers ($\theta_3, \theta_5$):** For layers updated by both devices, the server computes a weighted aggregation of the updates from both devices.
>
>     - **Exclusive Layers ($\theta_1, \theta_2$):** For layers updated by a single device, the update is applied directly.
>
>     - **Unselected Layers ($\theta_4, \theta_6$):** Layers not selected by any device remain unchanged.
>
> This mechanism effectively accommodates structural heterogeneity, backed by a rigorous convergence analysis in Appendix D. We have included this concrete example in **Appendix B**.

---

> ### Author Response · Authors · 2025-11-21
> **Response to Reviewer Cret (2 / 3)**
>
> > __W3: Experimental Comparison.__
>
> - **Baseline Selection.** We respectfully clarify that our original baseline, FedRA [5] (2024 ECCV), represents the closest SOTA work (employing layer pruning for memory constraints), which FedPruner outperforms by **+5.57%** on LLaMA2-13B (Table 4).
>
> - **New Experiments.** To further validates our method's effectiveness, we benchmark FedPruner against three additional advanced pruning methods (see table below). FedPruner consistently outperforms all baselines, achieving average performance gains of up to 2.22%. This superiority stems from:
>
>   - **vs. DepthFL:** DepthFL creates submodels by depth, causing deep layers to be trained only by the few high-memory devices. This results in undertraining of deep layers. FedPruner ensures balanced training across layers via dynamic orchestration.
>   -  **vs. FedMef:** FedMef is optimized for CNNs, overlooking the fact that parameters are the dominant bottleneck for LLMs (Fig. 1(c)), and neglects the impact of data heterogeneity.
>   - **vs. ShortGPT:** ShortGPT prunes layers based on block influence scores. This approach overlooks the functional specificity and collaborative dependencies between layers, which our macro-micro synergistic pruning framework explicitly models.
>
>   Detailed results and discussions are added to **Appendix J.1**.
>
>   **Table: Performance comparison against advanced pruning methods on TinyLLaMA**.
>
>   | **Method**               | **TruthfulQA** | **MMLU** | **IFEval** | **BBH** | Average             |
>   | ------------------------ | -------------- | -------- | ---------- | ------- | ------------------- |
>   | DepthFL [6] (2023, ICLR) | 37.72%         | 24.85%   | 16.14%     | 26.00%  | __26.18% (-2.15%)__ |
>   | FedMef [7] (2024, CVPR)  | 37.68%         | 24.79%   | 16.05%     | 25.93%  | __26.11% (-2.22%)__ |
>   | ShortGPT [8] (2025, ACL) | 37.91%         | 25.00%   | 16.18%     | 26.10%  | __26.30% (-2.03%)__ |
>   | **FedPruner**            | 39.16%         | 26.18%   | 19.87%     | 28.09%  | __28.33%__          |
>
>
>
> > **W4: Concerns on Experimental Breadth.**
>
> Thanks for your constructive suggestion. We agree that demonstrating robustness on more recent models is crucial for validating the relevance and scalability of our findings.
>
> - **Validation on SOTA LLMs:** To thoroughly address this, we extend our evaluation to include LLaMA3.2-3B, LLaMA3.1-8B, and Qwen2.5-7B. The results presented in the table below confirm that FedPruner significantly and consistently outperforms the theoretical (LoRA) baseline, achieving substantial average gains of **+8.62%, +8.11%, and +5.15%**, respectively. These results demonstrate the robustness of our method across diverse architectures and have been incorporated into **Section 4.9**.
> - **Future Directions:** We concur that extending FedPruner to generative and multimodal domains is pivotal. As FedPruner’s core mechanisms (FDLO and IALS) are architecture-agnostic, they are theoretically transferable to other Transformer-based backbones, including Diffusion models and MLLMs (e.g., Qwen-VL, LLaVA). We have further elaborated on this promising avenue in **Appendix N**.
>
> **Table: Performance evaluation on SOTA LLMs**.
>
> | Model           | Method        | TruthfulQA | MMLU   | IFEval | BBH    | Average             |
> | --------------- | ------------- | ---------- | ------ | ------ | ------ | ------------------- |
> | **LLaMA3.2-3B** | LoRA          | 44.25%     | 54.19% | 53.60% | 46.84% | __49.72%__          |
> |                 | **FedPruner** | 49.62%     | 60.43% | 66.58% | 56.71% | __58.34%(+8.62%)__  |
> | **LLaMA3.1-8B** | LoRA          | 48.07%     | 63.31% | 47.32% | 62.69% | __55.35%__          |
> |                 | **FedPruner** | 53.98%     | 67.95% | 61.40% | 70.51% | __63.46% (+8.11%)__ |
> | __Qwen2.5-7B__  | LoRA          | 48.02%     | 41.89% | 33.68% | 42.63% | **41.56%**         |
> |                 | **FedPruner** | 54.65%     | 46.70% | 39.51% | 45.96% | __46.71% (+5.15%)__ |

---

> ### Author Response · Authors · 2025-11-21
> **Response to Reviewer Cret (3 / 3)**
>
> > **W5: Missing Related Work and Contribution Positioning.**
>
> We apologize for the omission and have added a detailed discussion of these excellent works in **Section 5**. We wish to clarify that these works address different, and often secondary, bottlenecks in federated fine-tuning. In contrast, FedPruner is designed to solve the **primary and most prohibitive bottleneck: model parameters**.
>
> - **vs. FlexLoRA[9] & LEGENDS[10]:** These methods focus on heterogeneous LoRA adaptation.
>   - **Our Rationale:** As detailed in Fig. 1(c), optimizing LoRA modules provides very limited memory savings because they constitute a tiny fraction of the total memory (e.g., only 0.018% for LLaMA2-7B).
>   - **Memory Bottleneck:** The fundamental "memory wall" is the model parameters (accounting for 92.8% of memory), which these methods do not address.
>
>   - **Contribution Positioning:** Crucially, these approaches still presume the device can load the full model into memory. Our method removes this prerequisite entirely by intelligently pruning the global model. Moreover, our work is complementary: one could apply FlexLoRA or LEGENDS on top of the submodel created by FedPruner to achieve further LoRA optimization.
>
> - **vs. FedMef:** This method is optimized for CNNs, where activations are the main bottleneck.
>   - **Our Rationale:** This core assumption does not hold true for LLMs, where parameters are the dominant memory bottleneck. Moreover, FedMef employs unstructured weight pruning. In contrast, our structured layer pruning offers more direct, predictable, and hardware-agnostic advantages. Unstructured pruning often requires specialized libraries to realize actual memory or speed benefits, making it less practical for heterogeneous edge devices in an FL system.
>
>   - **Contribution Positioning:** This work is orthogonal to our approach, as it targets different components (individual weights and activations).
>
> **Empirical Validation**. To empirically validate the superiority of FedPruner, we conduct new experiments comparing FedPruner against these methods on TinyLLaMA. The results confirm FedPruner's effectiveness, achieving up to 2.22% average performance improvement. This is because the memory savings from these methods are minimal for LLMs. They still cannot effectively include the resource-constrained devices, which ultimately damages the final model's performance by excluding their valuable data. In contrast, FedPruner enables 100% device participation, translating directly into significant performance gains. Relevant experiments have been added to **Appendix J.3**.
>
> **Table: Performance comparison with other related works on TinyLLaMA**.
>
> | **Method**    | **TruthfulQA** | **MMLU** | **IFEval** | **BBH** | Average             |
> | ------------- | -------------- | -------- | ---------- | ------- | ------------------- |
> | FedMef [7]    | 37.68%         | 24.79%   | 16.05%     | 25.93%  | __26.11% (-2.22%)__ |
> | FlexLoRA [9]  | 38.52%         | 25.41%   | 17.62%     | 27.26%  | __27.20% (-1.13%)__ |
> | LEGENDS [10]  | 38.58%         | 25.53%   | 17.81%     | 27.42%  | __27.34% (-0.99%)__ |
> | **FedPruner** | 39.16%         | 26.18%   | 19.87%     | 28.09%  | __28.33%__          |
>
>
>
> ---
>
>
>
> Again, we thank the reviewer for the valuable feedback. Please let us know if there are any other questions or suggestions.
>
>
>
> __References:__
>
> [1] FwdLLM: Efficient Federated Finetuning of Large Language Models with Perturbed Inferences. 2024 ATC.
>
> [2] Flora: Federated fine-tuning large language models with heterogeneous low-rank adaptations. 2024 NeurIPS.
>
> [3] Slimgpt: Layer-wise structured pruning for large language models. 2024 NeurIPS.
>
> [4] LaCo: Large Language Model Pruning via Layer Collapse. 2024 Arxiv.
>
> [5] Fedra: A random allocation strategy for federated tuning to unleash the power of heterogeneous clients. 2024 ECCV.
>
> [6] Depthfl: Depthwise federated learning for heterogeneous clients. 2023, ICLR.
>
> [7] Fedmef: Towards memory-efficient federated dynamic pruning. 2024, CVPR.
>
> [8] Shortgpt: Layers in large language models are more redundant than you expect. 2025, ACL.
>
> [9] Federated Fine-tuning of Large Language Models under Heterogeneous Tasks and Client Resources. 2024, NeurIPS.
>
> [10] Adaptive Parameter-Efficient Federated Fine-Tuning on Heterogeneous Devices. 2025, IEEE TMC.

---

> ### Author Response · Authors · 2025-11-27
>
> Dear Reviewer Cret,
>
> As the discussion phase is entering its final stage, we want to kindly follow up to ensure that our response has adequately addressed your concerns.
>
> We truly value the time you have dedicated to reviewing our work. If there are any remaining questions or if further clarification is needed, please let us know—we are eager to engage in further discussion to improve the paper.
>
> Thank you again for your constructive feedback.
>
> Best regards,
>
> The Authors

---

### Author Response · Authors · 2025-11-30
**TL;DR A Summary of Discussion by Authors**

**Dear Reviewers, AC, and Researchers,**

**Thank you all for your dedicated effort at this difficult time to our ICLR community**. We are grateful for the constructive feedback from four reviewers, which has significantly helped us improve the quality and rigor of our manuscript. In the revised PDF, we have supplemented substantial experiments and clarifications to strengthen our work. We have detailed all extension experiments conducted during the rebuttal period in Section 4.9, Section 5, and Appendices J-N, which effectively address the reviewers' concerns.

We first recap the major concerns raised by reviewers and our corresponding solutions:

- **[Feasibility & Logical Grounding]:** Addressed concerns regarding the feasibility of calculating similarity on edge devices, the rationale for targeting model parameters, and the formal "memory wall" formulation. (W1@Cret, W1&2@qN12, W1-3@Lk6t)
- **[Baselines & SOTA Models]:** Addressed the need for comparisons against more recent baselines and validation on modern SOTA LLMs. (W3&4&5@Cret, W3&4@qN12, W4@Lk6t)
- **[FL Settings & Scalability]:** Addressed concerns regarding scalability, adaptability to heterogeneity, and aggregation strategies. (W2@Cret, W5@qN12, W6@Lk6t, W1@nDop)

Now we summarize our discussion and the specific actions taken for each reviewer below:

- **Reviewer Cret**
  - **W1:** We clarified the memory bottleneck analysis (Fig. 1) and the performance discrepancy (Section 2.2), while specifying that the "10-layer pruning" served as a motivational stress test, to address the concern about logical grounding and experimental details.
  - **W2:** We detailed the client-specific submodel construction and per-layer aggregation mechanism (Appendix B) to illustrate how FedPruner handles structural heterogeneity and demonstrate its FL-specific design.
  - **W3&5:** We expanded the discussion of related work (Section 5) and conducted new comparative experiments (Appendix J), to address the concern regarding missing baselines and experimental adequacy.
  - **W4:** We extended our evaluation to LLaMA3.2-3B, LLaMA3.1-8B, and Qwen2.5-7B (Section 4.9), confirming FedPruner's model generalization capability, while also elaborating on its potential for MLLMs and diffusion models in Appendix N.
- **Reviewer qN12**
  - **W1:** We clarified the "Pipeline Inference" mechanism (Appendix L), explaining how devices calculate layer similarity using adaptive chunk loading, effectively resolving the memory contradiction concern.
  - **W2:** We provided a quantitative analysis demonstrating that the computational overhead of the initialization setup is negligible (Appendix K), to address the concern regarding additional resource consumption.
  - **W3:** We added a direct comparison with DepthFL while elucidating the inherent limitations of conventional approaches in the context of LLMs (Appendix K), to address the concern regarding missing traditional baselines.
  - **W4:** We extended our evaluation to LLaMA3.2-3B, LLaMA3.1-8B, and Qwen2.5-7B (Section 4.9), demonstrating FedPruner's strong model generalization capability.
  - **W5:** We performed comprehensive tests across different client populations and participation rates (Appendix J.4, J.5), validating FedPruner's robustness and scalability.
- **Reviewer Lk6t**
  - **W1-3:** We refined the memory profiling details (Fig. 1) and formalized the memory consumption model (Section 1, Eq. 1), elucidating that targeting $M_{params}$ addresses the critical "memory floor" bottleneck that precludes edge device participation.
  - **W4:** We added detailed discussions and comparisons against more related works (Appendix J.2), to address the concern regarding missing baselines.
  - **W5:** We clarified that PPL was used strictly for preliminary diagnosis, while our main evaluation relies on Accuracy and GPT-4 Scoring for downstream tasks.
  - **W6:** We provided a concrete example and mathematical formulation (Appendix B) of the per-layer aggregation strategy, to address the concern regarding support for heterogeneous resource settings.
- **Reviewer nDop**
  - **W1/Q1:** We provided the analytical formula in Appendix M (Eq. 20) for determining the number of layers ($K$) based on device memory budgets, to address the concern regarding the quantitative determination mechanism.
  - **W2:** We clarified that our main evaluation relies on Accuracy and GPT-4 Scoring for downstream tasks (Section 4.1), to address the concern regarding metric ambiguity.
  - **W3/Q3:** We clarified the workflow in Algorithm 1, stating that the server distributes all LoRA parameters while clients only upload updates for their specific submodel, to address the concern regarding the communication process.
  - **Q2:** We explained the stochastic nature of the Importance-Aware Layer Selection, where layers with similar functionality receive similar probabilities, to address the concern regarding specific layer selection outcomes.

**FedPruner Authors**

---

### Meta-Review · Area_Chair_YCWo · 2026-01-04

**Summary:**

This paper proposed FedPruner, layer-wise pruning to reduce memory cost in federated learning. This paper received borderline scores (4, 4, 4, 6). Authors provided additional experimental results in the rebuttal, unfortunately the general engagement level in discussion is low, and there is no indication that reviewers are satisfied with the rebuttal. Reviewers raised several concerns: use old Llama-2 models; inaccurate claims on memory usage; unclear method description on pruning and aggregation; missing related work etc. Most of the clarification questions are reasonably addressed, and additional experiments on Llama-3 and Qwen2.5 are provided. However, the main concern on positioning the contribution in the literature appears to be unsolved. I would encourage the authors to incorporate the results in discussion to the paper, and further improve the presentation to clarify contributions.

**Reviewer Concerns:**

Reviewers raised several concerns: use old Llama-2 models; inaccurate claims on memory usage; unclear method description on pruning and aggregation; missing related work etc. Most of the clarification questions are reasonably addressed, and additional experiments on Llama-3 and Qwen2.5 are provided. However, the main concern on positioning the contribution in the literature appears to be unsolved.

**Reviewer Scores:**

This paper received borderline scores (4, 4, 4, 6). I think the authors did a reasonable good job in discussions, which should lead to score raising of 1-2 in average. But unfortunately the general engagement level in discussion is low, and there is no indication that reviewers are satisfied with the rebuttal

---

### Decision · Program_Chairs · 2026-01-26

Reject